# Modeling the effects of sediment concentration on the propagation of flash floods in an Andean watershed

María Teresa Contreras[1,2,3] and Cristián Escauriaza[1,2]

[1]Departamento de Ingeniería Hidráulica y Ambiental, Pontificia Universidad Católica de Chile. Av. Vicuña Mackenna 4860, 7820436, Santiago, Chile.
[2]Centro de Investigación para la Gestión Integrada de Desastres Naturales (CIGIDEN), Chile.
[3]Department of Civil and Environmental Engineering and Earth Sciences, University of Notre Dame, IN, 46556, USA

**Correspondence:** Cristián Escauriaza (cescauri@ing.puc.cl)

**Abstract.** Rain-induced flash floods are common events in regions near mountain ranges. In peri-urban areas near the Andes the combined effects of the changing climate and ENSO have resulted in an alarming proximity of populated areas to flood-prone streams, increasing the risk for cities and infrastructure. Simulations of rapid floods in these watersheds are particularly challenging, due to the complex morphology, the insufficient hydrometeorological data, and the uncertainty posed by the variability of sediment concentration. High concentrations produced by hillslope erosion and rilling by the overland flow in areas with steep slopes and low vegetational covering can change significantly the dynamics of the flow as the flood propagates in the channel. In this investigation, we develop a two-dimensional finite-volume numerical model of the non-linear shallow water equations coupled with the mass conservation of sediment to study the effects of different densities, which include a modified version of the quadratic stress model to quantify the changes on the flow rheology. We carry out simulations to evaluate the effects of the sediment concentration on the floods in the Quebrada de Ramón watershed, a peri-urban Andean basin in central Chile. We simulate a confluence and a total length of the channel of 10.4 km, with the same water hydrographs and different combinations of sediment concentrations in the tributaries. Our results show that the sediment concentration has strong impacts on flow velocities and water depths. Compared to clear water flow, the wave-front velocity slows down more than 70% for floods with a volumetric concentration of 60% and the total flooded area is 36% larger when the sediment concentration is equal to 20%. The maximum flow momentum at cross-sections in the urban area increases 14.5% on average when the mean concentration along the main channel changes from 30% to 44%. Simulations also show that other variables such as the arrival time of the peak flow, and the shape of the hydrograph at different locations along the channel are not significantly affected by the sediment concentration and depend mostly on the steep channel morphology. Through this work we provide a framework for future studies aimed at improving hazard assessment, urban planning, and early warning systems in urban areas near mountain streams with limited data and affected by rapid flood events.

# 1 Introduction

Flash floods with high sediment concentrations are common natural events in mountain rivers, which generate hazards in cities and other smaller human communities located near river channels (European Environmental Agency (EEA), 2005; Wilby et al., 2008). In spite of the continued efforts to provide structural and non-structural measures to control flood hazards in general, economical losses have increased in recent decades (Slater et al., 2015), and flood risks and vulnerability associated with various economic, political, and social processes are also expected to increase in the future (Pelling, 2003; Blaikie et al., 2004; Bankoff et al., 2004).

The spatial and temporal distribution of precipitation, the morphology of the basin, soil properties, and vegetation characteristics, naturally influence the magnitude and frequency of floods and sediment transport. Anthropogenic factors also affect the volume and peak discharges of floods in mountain rivers. Climate models predict a larger frequency of intense precipitation events and cyclonic weather systems that will increase the vulnerability in many mountainous regions in the future (Sanders, 2007; Arnell and Gosling, 2014; Boers et al., 2014). An amplification of the flood hazards is also expected due to the continuing expansion of cities located in floodplains (Jongman et al., 2012; Hirabayashi et al., 2013), accelerated urbanization processes (Schubert et al., 2008), lack of urban planning (Rugiero and Wyndham, 2013), and changes in land-use and cover (Kundzewicz et al., 2014).

The effectiveness to assess flood hazards and to design strategies aimed at reducing potential damages caused by flooding is closely related to understanding the dynamics of the flow in real conditions. Recent physical models and experiments have provided relevant insights on the flow physics of flash floods in extreme conditions (e.g. Testa et al., 2007). Field-based and experimental research over complex topography, however, require large facilities with advanced instrumentation to provide high-resolution measurements that are also limited by the spatio-temporal scales at which rapid floods occur. Numerical models, on the other hand, have also become fundamental tools to advance our understanding on the dynamics of floods, evaluating complex scenarios and predicting water depths and flow velocities in arbitrary geometries (Siviglia and Crosato, 2016). Simulations yield detailed information on the flood dynamics, which is sometimes experimentally inaccessible or cannot be directly measured in the field. They can also complement measurements, becoming effective tools for urban planning and for designing early warning systems during flood events (Mignot et al., 2005; Schubert and Sanders, 2012).

In most hydrodynamic models to simulate flood propagation, the nonlinear shallow water equations (NSWE) or Saint Venant equations are employed to describe the dynamics of the flow in homogeneous and incompressible fluids. They are obtained by vertically averaging the three-dimensional Navier-Stokes equations, assuming a hydrostatic pressure distribution, resulting in a set of horizontal two-dimensional (2D) hyperbolic conservation laws that describe the evolution of the water depth and depth-averaged velocities in space and time. In flows where discontinuities and rapid wet-dry interfaces develop, numerical models employ Godunov-type of formulations, solving a Riemann problem at the interfaces of the elements of the discretization (Anastasiou and Chan, 1997; Toro, 2001).

The development of efficient and accurate numerical models to simulate flash floods, however, is far from trivial, since multiple factors control the dynamics of the flow. Especially in mountainous regions, where rivers are characterized by three

important features that complicate their representation: (1) Complex bathymetries and steep slopes produce rapid changes on velocities and water depths, formation of bores, and wet-dry interfaces; (2) Large sediment concentrations affect directly the flow hydrodynamics by introducing additional stresses that alter the momentum balance of the instantaneous flow; and (3) Lack of accurate field data, used for validation, due to the difficulties on measuring hydrometeorological variables in high-altitude environments, with difficult access, and during episodes of severe weather.

The Andes mountains in South America incorporate all these characteristics, and they have been the scenario of many recent catastrophic events, leaving a significant human toll and economic losses (Wilcox et al., 2016). The region is characterized by rapid floods with high concentrations of sediment, generally produced by hillslope erosion and rilling by the overland flow in areas with steep slopes and low vegetational covering. Additional factors, such as the storms caused by the South-American monsoon (Zhou and Lau, 1998), and El Niño-Southern Oscillation (ENSO) can generate anomalous heavy rainfall (Holton et al., 1989; Díaz and Markgraf, 1992), producing a great volume of liquid precipitation and significant erosion and sediment transport in the flow.

High sediment concentrations during floods cause additional stresses produced by the increase of the density and viscosity of the water-sediment mixture. Models need to account for the internal stresses that emerge from the particle-flow and particle-particle interactions in the sediment-laden flow. These stresses transform the rheological behavior of the mixture, represented by additional terms of momentum transfer in the governing equations. A wide variety of rheological models have been proposed depending on the sediment properties and concentration (see for instance Bingham, 1922; Bagnold, 1954; O'Brien and Julien, 1985, among others). These approaches are based on empirical equations that have been estimated from laboratory studies (Parsons et al., 2001), or back-calibrated from past events (Naef et al., 2006).

The main objective of this investigation is to gain fundamental insights on the effects of high sediment concentrations on the propagation of floods in an Andean watershed. We develop a 2D finite-volume numerical model of the NSWE, building on the work of Guerra et al. (2014), which incorporates the effects of the sediment load on the dynamics of the flow over natural terrains and complex geometries. First we validate the sediment-coupling module by using three benchmark cases including an analytical solution, numerical simulations, and experiments. Then we carry out simulations of flows with different sediment concentrations in the two main tributaries of the *Quebrada de Ramón* watershed, located at the foothills of the Andes mountain range, to the east of Santiago, Chile, where part of the city occupies the lower section of the river basin. From the simulations we evaluate the effects of the sediment load on the evolution of the flow depth and velocity, and we link the hydrodynamic response of the river channel to the variations of sediment concentration. The analysis provides quantitative information of the hyperconcentrated flood propagation, including the changes on the total flooded area and momentum at cross-sections of the flow, among other parameters for flood hazard assessment.

The paper is organized as follows: The governing equations of the model and its implementation on the study area are presented in sections 2 and 3, respectively. In section 4, we study the consequences of high concentrations on the dynamics of the flow, the total momentum in cross-sections of the river, and local water depths and velocities. In section 5, we discuss the results in the context of the interactions between geomorphic controls of the flood propagation and the sediment concentration of the flow. Finally, in section 6 we summarize the findings of this investigation.

## 2 Modeling hyperconcentrated flows

### 2.1 Governing Equations

Rapid floods over the complex topography of mountainous regions are commonly affected by high sediment concentrations, which change the rheology of the flow. By assuming that the mixture preserves the Newtonian constitutive relation between stress and rate of strain, the NSWE equations can be modified to account for the heterogeneous density distribution in space and time (Loose et al., 2005; Michoski et al., 2013).

The NSWE model implemented in this investigation has the following assumptions: (i) hydrostatic pressure distribution; (ii) negligible vertical velocities; (iii) vertically-averaged horizontal velocities; (iv) horizontal heterogeneous fluid density; (v) homogeneous density in the vertical direction; and (vi) fixed bed (neither erosion nor deposition). The momentum sources and sinks consider the gravity term, the bed resistance, and rheology of the mixture, including the yield stress, Mohr-Coulomb, viscous stresses, and turbulent and dispersive stresses, as discussed in section 2.2.

If we denote the dimensional variables of the flow with a hat (ˆ), the NSWE coupled with the sediment concentration are written as follows,

$$\frac{\partial \hat{\rho}\hat{h}}{\partial \hat{t}} + \frac{\partial \hat{\rho}\hat{h}\hat{u}}{\partial \hat{x}} + \frac{\partial \hat{\rho}\hat{h}\hat{v}}{\partial \hat{y}} = 0 \tag{1}$$

$$\frac{\partial \hat{\rho}\hat{h}\hat{u}}{\partial \hat{t}} + \frac{\partial}{\partial \hat{x}}\left(\hat{\rho}\hat{u}^2\hat{h} + \frac{1}{2}\hat{\rho}g\hat{h}^2\right) + \frac{\partial \hat{\rho}\hat{u}\hat{v}\hat{h}}{\partial \hat{y}} = -\hat{\rho}g\hat{h}\frac{\partial \hat{z}}{\partial \hat{x}} - \hat{\tau}_{\hat{x}} \tag{2}$$

$$\frac{\partial \hat{\rho}\hat{h}\hat{v}}{\partial \hat{t}} + \frac{\partial \hat{\rho}\hat{u}\hat{v}\hat{h}}{\partial \hat{x}} + \frac{\partial}{\partial \hat{y}}\left(\hat{\rho}\hat{v}^2\hat{h} + \frac{1}{2}\hat{\rho}g\hat{h}^2\right) = -\hat{\rho}g\hat{h}\frac{\partial \hat{z}}{\partial \hat{y}} - \hat{\tau}_{\hat{y}} \tag{3}$$

$$\frac{\partial C\hat{h}}{\partial \hat{t}} + \frac{\partial C\hat{h}\hat{u}}{\partial \hat{x}} + \frac{\partial C\hat{h}\hat{v}}{\partial \hat{y}} = 0 \tag{4}$$

where $\hat{h}$ is the flow depth, and $\hat{u}$ and $\hat{v}$ are the depth-averaged velocities in the cartesian coordinate directions $\hat{x}$ and $\hat{y}$, respectively. The bed elevation is denoted as $\hat{z}$, $g$ is the acceleration of gravity, $\hat{t}$ represents the time, $\hat{\rho}$ is the density of the water-sediment mixture, $C$ is the volumetric concentration of sediment, and $\hat{\tau}_x$ and $\hat{\tau}_y$ are the total stresses.

Here we follow the same procedure outlined in Guerra et al. (2014), expressing the governing equations in non-dimensional form using a characteristic velocity scale $\mathcal{U}$, a scale for the water depth $\mathcal{H}$, and a horizontal length scale of the flow $\mathcal{L}$. In this case, two non-dimensional parameters appear in the equations, i.e. the relative density between the sediment and water $s = \rho_s/\rho_w$, and the Froude number $Fr = \mathcal{U}/\sqrt{g\mathcal{H}}$.

To adapt the computational domain to the complex arbitrary topography in mountainous watersheds, we use a boundary fitted curvilinear coordinate system, denoted by the coordinates $(\xi, \eta)$. Through this transformation we can have a better resolution

in zones of interest and an accurate representation of the boundaries. We perform a partial transformation of the equations, and write the set of dimensionless equations in vector form as follows,

$$\frac{\partial Q}{\partial t} + J\frac{\partial F}{\partial \xi} + J\frac{\partial G}{\partial \eta} = S_b(Q) + S_S(Q) + S_C(Q) \tag{5}$$

where $Q$ is the vector that contains the non-dimensional cartesian components of the conservative variables $h$, $hu$, $hv$ and $hC$,

which are obtained by replacing the density of the mixture $\hat{\rho} = C\rho_s + (1-C)\rho_w$, where $\rho_w$ is the water density, and $\rho_s$ the sediment density.

The Jacobian of the coordinate transformation $J$ is expressed in terms of the metrics $\xi_x$, $\xi_y$, $\eta_x$ and $\eta_y$, such that $J = \xi_x\eta_y - \xi_y\eta_x$ (Lackey and Sotiropoulos, 2005; Guerra et al., 2014). The fluxes $F$ and $G$ in each coordinate direction expressed as follows,

$$F = \frac{1}{J}\begin{pmatrix} hU^1 \\ uhU^1 + \frac{1}{2Fr^2}h^2\xi_x \\ vhU^1 + \frac{1}{2Fr^2}h^2\xi_y \\ ChU^1 \end{pmatrix}, G = \frac{1}{J}\begin{pmatrix} hU^2 \\ uhU^2 + \frac{1}{2Fr^2}h^2\eta_x \\ vhU^2 + \frac{1}{2Fr^2}h^2\eta_y \\ ChU^2 \end{pmatrix} \tag{6}$$

where $U^1$ y $U^2$ represent the contravariant velocity components defined as $U^1 = u\xi_x + v\xi_y$ and $U^2 = u\eta_x + v\eta_y$, respectively.

The model considers three source terms: $S_B$ contains the bed slope terms, $S_S$ corresponds to the bed and internal stresses of the flow, and $S_C$ incorporates the effects of the spatial gradients of sediment concentration. This last term might be important in rapid flows with large concentration gradients, such as dam-breaks with sediment-laden debris flows (Cao et al., 2004), and

in cases with interactions of clear water and hyperconcentrated flows. The source vectors are expressed as follows,

$$S_b(Q) = \begin{pmatrix} 0 \\ \frac{-h(z_\xi\xi_x + z_\eta\eta_x)}{Fr^2} \\ \frac{-h(z_\xi\xi_y + z_\eta\eta_y)}{Fr^2} \\ 0 \end{pmatrix}; \; S_S(Q) = \begin{pmatrix} 0 \\ -S_x \\ -S_y \\ 0 \end{pmatrix}; \; S_C(Q) = \begin{pmatrix} 0 \\ \frac{-h^2(C_\xi\xi_x + C_\eta\eta_x)}{2Fr^2}\left(\frac{s-1}{C(s-1)+1}\right) \\ \frac{-h^2(C_\xi\xi_y + C_\eta\eta_y)}{2Fr^2}\left(\frac{s-1}{C(s-1)+1}\right) \\ 0 \end{pmatrix} \tag{7}$$

Since the objective of this investigation is to study exclusively the impacts of sediment transport on the hydrodynamics of rapid floods in mountain rivers, where the geomorphic features of the channel play a significant role on flood propagation, we are not considering the erosion or deposition of the bed. The channel of the *Quebrada de Ramón* stream has a long bedrock

section, and the urban area is completely paved. No significant erosion of the channel was reported in the most recent flood, but these conditions cannot apply to other similar cases (e.g. Wilcox et al., 2016).

It is important to note that mountain rivers with steep slopes in peri-urban watersheds exhibit a wide variety of bed conditions, i.e. bedrock channels, boulders, coarse gravel surfaces, armoring, sand and gravel mixtures, and the concrete surface of the urban setting. The complexity of these environments, and the unknown effects of the high sediment concentrations that originate

in the high-altitude sections of the mountains, prompted the development of this model that couples the transport equation to the flow in mass and momentum with high resolution. The system of governing equations 5 is solved using a well-balanced

second-order finite-volume method with an efficient Riemann solver that incorporates hydrostatic reconstruction, and a semi-implicit fractional-step time integration approach (Guerra et al., 2014) (see Appendix A). Here we incorporate the density and rheological models that are described in the following section.

## 2.2 Rheological Model

The classification and rheology of gravity-driven flows with higher concentrations usually depend on the particle size distribution and sediment composition. Depending on these characteristics, the flows can vary from nearly dry landslides to water flow, with intermediate conditions such as debris flows, mudflows, and mud floods (Julien and León, 2000; Naef et al., 2006). The rheological behavior that determines the magnitude of the momentum losses is incorporated in additional source terms of the hydrodynamic model previously presented in vector $S_S$ in equation 7. As summarized by Ancey (2007), gravity-driven

flows can be described by different rheological models such as Bagnold, Bingham, Voellmy, or Coulomb, depending on the assumptions of the effects of the particles on the dynamics of the flow. In our numerical model we implement the quadratic shear stress model developed by O'Brien and Julien (1985) (see also O'Brien et al., 1993), which represents the total stress $\hat{\tau}_i$ in each coordinate direction $i$, as follows,

$$\hat{\tau}_i = \hat{\tau}_{yield} + \hat{\mu}_m \frac{\partial \hat{u}_i}{\partial \hat{z}} + \hat{\zeta} \left( \frac{\partial \hat{u}_i}{\partial \hat{z}} \right)^2 \tag{8}$$

where $\hat{\tau}_{yield}$ represents the sum of the Mohr-Coulomb and yield stresses, the second term is the viscous shear-stress that depends on the dynamic viscosity of the mixture $\hat{\mu}_m$ and the vertical velocity gradient expressed as a function of the cartesian velocity components $u_i$. The last term corresponds to the sum of the turbulent and dispersive stresses, which depend quadratically of the velocity gradient and the inertial shear stress coefficient $\hat{\zeta}$, defined by the following equation,

$$\hat{\zeta} = \hat{\rho} \, \hat{l}_m{}^2 + c_{Bd} \, \rho_s \, \lambda^2 \, d_s^2 \tag{9}$$

where $\hat{l}_m = 0.4\hat{h}$ is the Prandtl mixing-length Julien and León (2000), $c_{Bd}$ is an empirical proportionality constant equal to $0.01$ according to Bagnold (1954), $d_s$ is the median sediment diameter, and $\lambda$ is Bagnold's linear concentration, which corresponds to the ratio between the grain diameter and the mean free dispersion distance. The magnitude of $\lambda$ is related to the volumetric concentration of the mixture and the maximum volumetric static concentration $C^*$, as defined by Bagnold (1954),

$$\lambda^{-1} = \left[ \left( \frac{C^*}{C} \right)^{\frac{1}{3}} - 1 \right] \tag{10}$$

In the present numerical model we modify the quadratic model of O'Brien and Julien (1985) to represent the stresses for a wide range of sediment concentrations, expressing clearly the contribution of each physical mechanism as the combination of relations that account for the stresses, which have been obtained from experiments or physically-based formulas. To determine the values of the source terms defined as $S_i = \tau_i/\rho$ in equation 7, the stresses are non-dimensionalized, depth-integrated, and added in the source term vector for each coordinate direction, such that the total stresses are expressed as:

$S_i = S_{yield} + S_{v_i} + S_{td_i}$                 (11)

where $S_{yield}$ represents the sum of the yield and Mohr-Coulomb stress, $S_{v_i}$ the viscous stress and $S_{td_i}$ the sum of the dispersive and turbulent stresses. Each of these terms are computed separately from empirical formulas.

The yield and Mohr-Coulomb stresses $S_{yield}$ are jointly calculated from the following expression:

$$S_{yield} = \frac{\mathcal{L}}{\mathcal{H}} \left[ \frac{\tau_{yield}}{\rho} \right] \tag{12}$$

in which the yield shear stress and the density of the mixture are non-dimensionalized with the scale of the inertia $\rho_w \mathcal{U}^2$, and the water density $\rho_w$, respectively. The yield stress is isotropic and calculated using the following empirical relation given in SI units:

$$\hat{\tau}_{yield} = a\, 10^{bC} \tag{13}$$

where for typical soils, the experimental coefficients $a$ and $b$ are equal to 0.005 and 7.5, respectively (Julien, 2010).

The viscous term $S_{v_i}$ is computed from the bed stress in each direction $\tau_x = \rho\, C_f\, u\, \sqrt{u^2 + v^2}$ and $\tau_y = \rho\, C_f\, v\, \sqrt{u^2 + v^2}$, using the laminar friction coefficient defined as $C_f = k/Re$, where $k$ is the viscous resistance parameter equal to $64$ in open-channel flows (Sturm, 2001). The Reynolds number is defined as $Re = \frac{\rho \sqrt{u^2 + v^2} h}{\mu_m}$, where the dynamic viscosity of the mixture is non-dimensionalized as $\mu_m = \frac{\hat{\mu}_m}{\rho_w \mathcal{U} \mathcal{H}}$. Thus, the expression used to represent the viscous losses is written as follows:

$$S_{v_i} = \frac{\mathcal{L}}{\mathcal{H}} \left[ \frac{k\, \mu_m\, u_i}{8\rho h} \right] \tag{14}$$

To estimate $\mu_m$ we use the formula proposed by Eyring (1964) and Thomas (1965). This relation is a function of the volumetric sediment concentration in the mixture and the dynamic viscosity of water $\mu_w$ in SI Units:

$$\frac{\hat{\mu}_m}{\mu_w} = 1 + 2.5C + 10.05C^2 + 0.00273 \exp\left(16.6C\right) \tag{15}$$

To compute the last term in equation 11, $S_{td}$, a Manning or Chézy coefficient is used to represent the friction factor $C_f$ in the bed stress formula, resulting in the following expression for each cartesian coordinate direction:

$$S_{td_i} = \begin{cases} \text{Manning:} & \frac{\mathcal{L}}{\mathcal{H}} \left[ \frac{n_{td}^2\, u_i\, \sqrt{u^2 + v^2}}{Fr^2\, h^{1/3}} \right] \\ \text{Chézy:} & \frac{\mathcal{L}}{\mathcal{H}} \left[ \frac{u_i\, \sqrt{u^2 + v^2}}{Fr^2\, C_{z_{td}}^2} \right] \end{cases} \tag{16}$$

Since $S_{td}$ represents the sum of friction, turbulence and dispersive stresses, we use either a modified Manning $n_{td}$ or Chézy $Cz_{td}$ coefficients. To estimate their value we add two Darcy-Weisbach friction factors, denoted as $f_t$ and $f_d$, representing the turbulent and dispersive effects respectively. To compute $f_t$, we use Colebrook's equation:

$$\frac{1}{\sqrt{f_t}} = -2\log\left( \frac{\hat{k}_s}{3.7\mathcal{H}h} + \frac{2.51}{Re\sqrt{f_t}} \right) \tag{17}$$

The value of $f_t$ is calculated as a function of the depth of the mixture $h$, the Reynolds number, and the bed specific roughness $\hat{k}_s$, which is estimated as follows (Bathurst, 1978),

$$\hat{k}_s = 6.8 d_s \ \ \text{[SI Units]} \tag{18}$$

To account for the dispersive effects, $f_d$ is calculated using the relation proposed by Takahashi (2007):

$$\sqrt{\frac{8}{f_d}} = \frac{2\mathcal{H}h}{5d_s} \left\{ \frac{1}{0.02} \left[ C + (1-C)\frac{\rho_w}{\rho_s} \right] \right\}^{\frac{1}{2}} \lambda^{-1} \tag{19}$$

where $\rho_s$ and $\rho_w$ are the sediment and water densities, respectively, and $\lambda$ is Bagnold's linear concentration defined previously in equation 10. In general, numerical simulations show that the turbulent friction coefficient $f_t$ is significantly smaller than the dispersive factor $f_d$ (D'Aniello et al., 2015). Dispersive effects, however, become important for low values of relative roughness ($\frac{h}{d_s} < 50$), as discussed in detail by Julien and Paris (2010).

To obtain the terms $S_{td_i}$, we use the following relation proposed by Julien (2010), to transform the combined Darcy-Weisbach friction coefficient $f_{td} = f_t + f_d$ in an equivalent Manning o Chézy coefficient:

$$\sqrt{\frac{8}{f_{td}}} = C_{z_{td}} Fr = \frac{h^{1/6} Fr}{n_{td}} \tag{20}$$

Appendix B contains tests of this model comparing to analytical solutions, numerical simulations, and experiments, to verify its precision and evaluate the flexibility of the model to address the flood propagation in Andean environments.

## 3   Study case: Floods in the Quebrada de Ramón

As it has been previously mentioned, one of the cases where the sediment concentration plays a significant role is in the flood propagation in mountain rivers. We select as a study case the Quebrada de Ramón watershed in the Andes of central Chile to evaluate how the hydrodynamics of the flow is altered by the magnitude of the sediment concentrations. We simulate the same scenario, but modify the volume sediment concentration in a range 0-60%. We characterize locally the depths and velocities of the flow as the flood propagates, but we also evaluate the extension of flooded area and the momentum of the flow in the lower section of the watershed, within the city of Santiago.

In this watershed there are two major tributaries draining the north and south sections of the watershed. We define the computational domain shown in Figure 1, which comprises a total distance of 10.4 km along the main channel. The highest part of the computational domain is located at an elevation of 2,212 m asl, with the Quebrada de Ramón stream (QR) and the Quillayes stream (Qui) approaching from the north and south respectively. The upstream boundary of the domain is located 3 km upstream of the confluence (Figure 2c). The channel downstream the confluence continues to the flood zone, with a single main river channel shown in Figure 2b, and ends at an elevation of 652 m asl, where the stream has been channelized in the city, as shown in Figure 2a. A curvilinear boundary-fitted grid is used to perform the simulations, consisting of a total of $10,070 \times 218$ grid nodes. The grid resolution varies progressively in the flow direction from 0.5 m upstream and near the confluence, to 2 m of resolution within the flooding zone. On the cross-stream direction, the mean resolution of the grid close to the channels is approximately 1 m. To construct the grid, we use a 1m resolution LIDAR of the area around the channels. The LIDAR data is coupled to a 30 m resolution digital elevation model (DEM) from satellite images for the rest of the watershed.

The bed roughness is represented by a mean sediment grain diameter $d_s$. Field measurements are used to interpolate the values of $d_s$ in the entire computational domain using the nearest neighbor method. The mean sediment grain size distribution

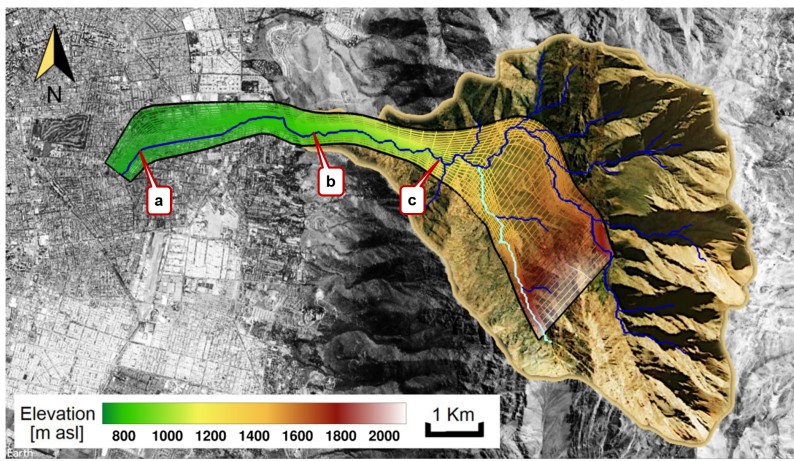

**Figure 1.** Satellite image of the Quebrada de Ramón watershed and the computational domain. The area enclosed by the black line is defined from the LIDAR topography, and incorporates the section of the city around the river channel. The main channel is depicted in blue and the tributary in turquoise.

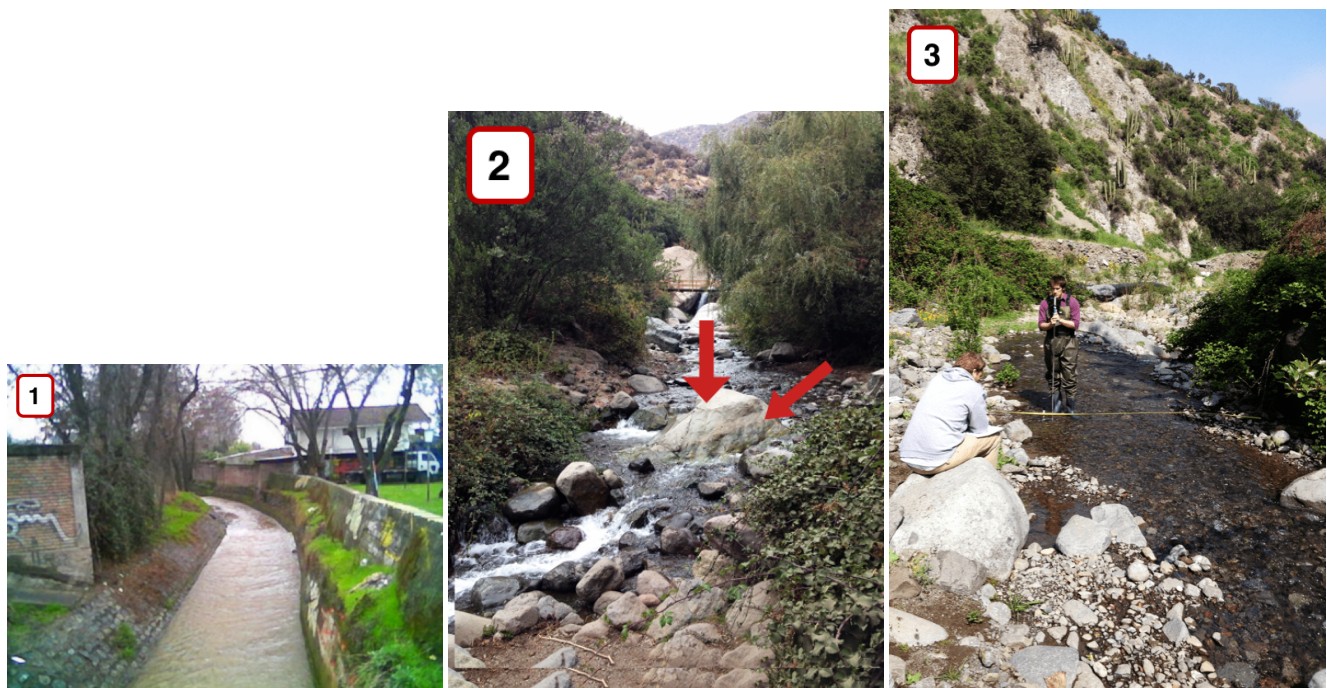

**Figure 2.** Three photos along the Quebrada de Ramón stream from downstream to upstream: 1) The channelized section, 2) The floodplain, and 3) The confluence of the Quebrada de Ramón and the Quillayes stream.

is shown in Figure 3, along with the 7 points where we report the dynamics of the flow, which include locations at the highest elevation of the domain (denoted as QR-U and Qui-U), upstream of the confluence (QR-D and Qui-D), downstream of the confluence (QR-C), the flooding zone (QR-F), and at the outlet (QR-E).

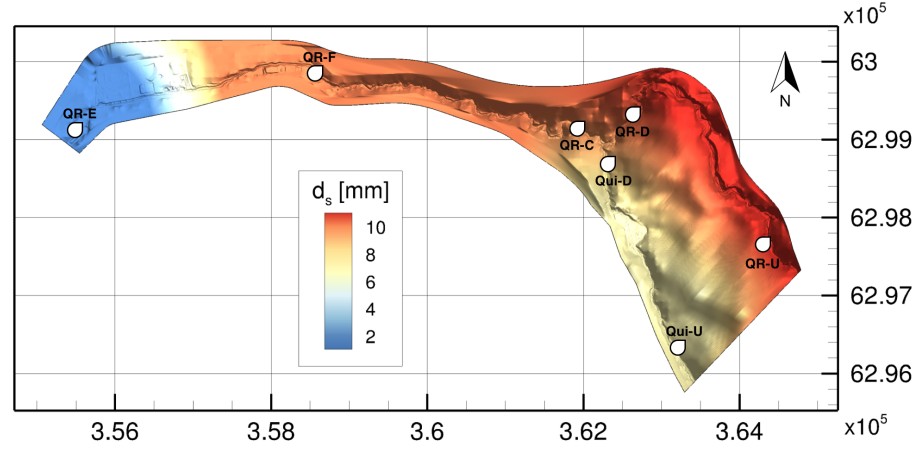

**Figure 3.** Distribution of the mean sediment size in millimeters. White circles denote the measurement sites where we report the propagation of the flood.

The hydrographs of the event studied in this investigation for the two main rivers that correspond to the tributaries of the confluence are shown in Figure 4. The return period of these hydrographs have been is estimated in 50 years, and they were obtained from a continuous semi-distributed hydrological model built in HEC-HMS, for the 1971 - 2010 period (Ríos, 2016). This case is selected since the peak flow at the outlet is expected to exceed significantly the capacity of the channelized section in the city.

The actual sediment concentration during flash floods in Andean watersheds is unknown in most cases. In addition to the lack of information in these rivers, landslides due to erosion produced by soil saturation in the steep slopes of the mountains are common. These conditions can increase considerably the sediment supply to the streams during flood events, with material that does not come from the channel but mostly from hillslope erosion. We study the dynamics of the flood for different scenarios by carrying out a series of simulations to compare and understand the flood hazards and effects of hyperconcentration on the two main streams of the Quebrada de Ramón watershed.

We simulate four different scenarios considering different concentrations of $0\%$, $20\%$, $40\%$ and $60\%$ equal in both streams. Two another cases are simulated, with concentrations of $20\%$ in the Quebrada de Ramón stream and $60\%$ in the Quillayes stream, and viceversa.

The river bed is considered dry at the beginning of the simulation, to avoid the additional effects of different sediment concentrations of the initial flow.

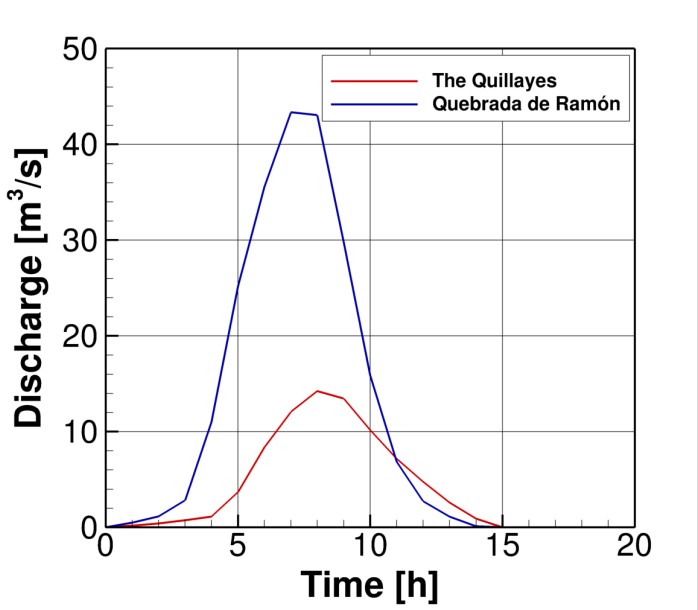

**Figure 4.** 50-year return period hydrographs in both streams obtained from a hydrological model of the catchment (Ríos, 2016). This conditions in the Quebrada de Ramón and at the Quillayes stream can potentially produce a large-scale flood in the watershed.

We perform the simulations for a total physical time of 1 day, using a simulation time step defined by the Courant-Fiedrichs-Lewy criterion (CFL) stability criterion, defined as:

$$CFL = \Delta t \times \frac{max\left[max\left(U^1 + \sqrt{gh}\sqrt{(\xi_x^2 + \xi_y^2)_{i,j}}\right), \left(U^2 + \sqrt{gh}\sqrt{(\eta_x^2 + \eta_y^2)}\right)_{i,j}\right]}{min\left(\Delta\xi, \Delta\eta\right)} \tag{21}$$

which in this case is set equal to $CFL = 0.9$.

5    The inflow boundary condition in Figure 4 is used at the eastward boundaries, and open boundary conditions are considered at all the other boundaries of the computational domain.

## 4   Results: Effects of the sediment concentration on the flood propagation

To evaluate the impacts of different sediment concentrations on the flood dynamics, in the following subsections we analyze the flow hydrodynamics including: (1) the position and velocity of the flood wave front; (2) the peak flow and arrival time, (3) 10  the flooded areas; (4) the effect of the sediment concentration on the depth and flow velocity; and (5) the momentum of the flow in the urban zone.

**Table 1.** Mean velocity of the front for different sediment concentrations. In parenthesis, the percentage change from clear water.

| Volumetric concentration [−] | Mean velocity of the wave front [km/h] | |
| :---: | :---: | :---: |
| | Quebrada de Ramón stream | Quillayes stream |
| 0% | 2.88 (0%) | 4.99 (0%) |
| 20% | 1.88 (34.73%) | 3.10 (37.88%) |
| 40% | 1.64 (43.06%) | 1.27 (74.55%) |
| 60% | 1.54 (46.53%) | 1.12 (77.56%) |

## 4.1  Position and mean velocity of the wave front

To quantify the propagation of the flood along the channels and the arrival time of the flood to the city, we compute the mean velocity of the wave front by tracking its position in time. Table 1 shows the mean velocity in the section upstream of the confluence, for the Quebrada de Ramón and Quillayes stream. As it can be anticipated, the velocity of the wave front decreases with the concentration, as interparticle collisions and internal stresses reduce the momentum of the flow, increasing flow resistance. Note that the flood propagation velocity is very sensitive to variations of concentration in more dilute conditions. Overall, the velocity for a concentration of 20% is 30% slower than clear water flow on both streams. On the other hand, when the concentration increases from 40% to 60%, the velocity is reduced by less than 10%. The mean wave front propagation velocity seems to be decreasing quadratically with the concentration in this case ($R^2 = 0.983$ and $R^2 = 0.9868$ for each stream).

Figure 5 shows the location of the wave front vs time for both streams upstream of the confluence. The inverse slopes of these curves represent the instantaneous velocity of the front for each sediment concentration we have simulated.

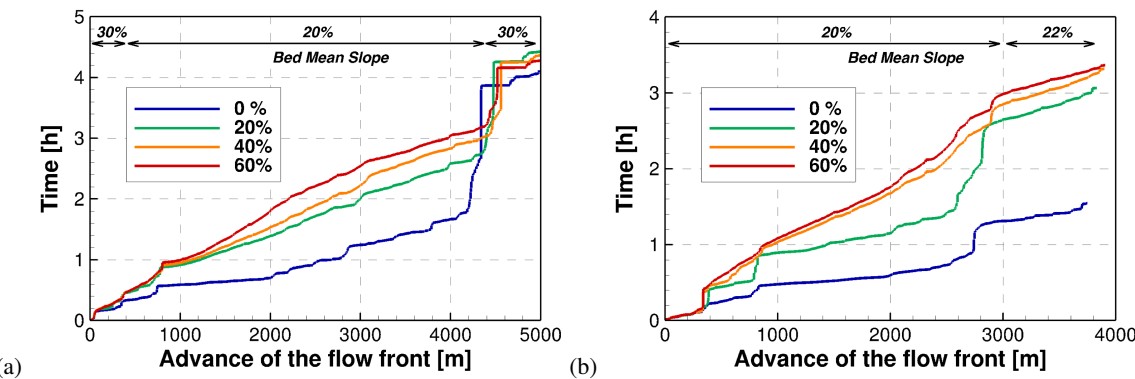

**Figure 5.** Location of the wave front in time for both streams, considering four different sediment concentrations, to characterize the advance of the flood in the river: (a) Quebrada de Ramón; and (b) The Quillayes stream.

The numerical results show that the sediment concentration produces a significant change on the evolution of the flood, as it is the only factor that we modify in these simulations. The local variations in these velocities are produced by the gradual change of bed roughness and the slope of the river channels, which is approximately constant in large portions of the reaches. The Quillayes stream exhibits higher propagation velocities, which are also consistent with the steeper slopes and finer sediment diameters on the bed.

Figure 5 shows that the flood propagation has deceleration stages of the front, seen as steps in the location of the front in time. Two clear steps are observed in the Quebrada de Ramón stream. The first is located at 800 m from the inflow boundary of the computational domain, which is produced by a local widening of the channel. The second deceleration, at 4,500 m, is generated by a narrowing of the river that accumulates a large volume and reduces the velocity of the flow, increasing the depth upstream of this section due to backwater effects. In the Quillayes stream, we observe three deceleration stages at 400, 800, and 2,800 m, which are also caused by local widening of the channel. These detailed dynamic features of the flood are modulated by the sediment concentration, as the hyperconcentrated cases show a more uniform propagation of the front.

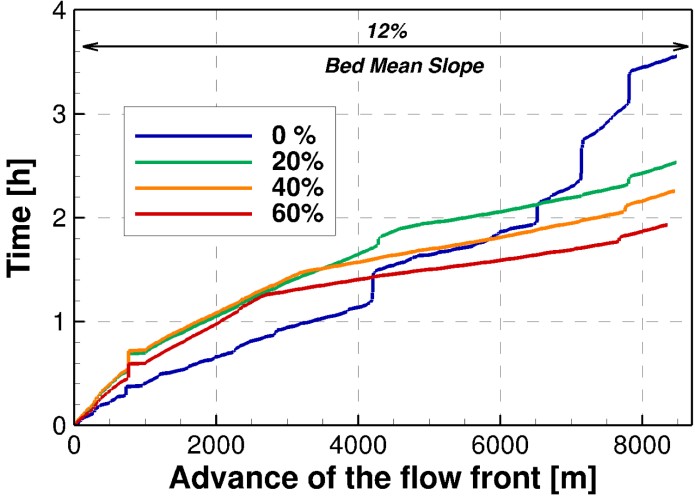

**Figure 6.** Location of the wave front in time for the Quebrada de Ramón downstream of the confluence considering four sediment concentrations in both streams.

Downstream of the confluence, we also observe the effects of the interaction between the morphology and the sediment concentration. The wave-front speed is affected by the different arrival times of flows from both tributaries. Figure 6 shows the distance traveled by the flood from the confluence to the outlet of the watershed. The origin is defined at the junction, and the time starts when the flood from the Quillayes, the faster and smaller stream, reaches the confluence. The time lapse between the arrivals is larger for flows with lower concentrations. This difference is equal to 3 hours in clear-water flows, but only 1 hour for a concentration of 60%, which changes the hydrodynamics of the wave-front when it arrives to the lower section of the channel, in the urban area. Along the first 4 km, we observe a similar dynamics to Figure 5, where the wave-front speed is

faster as the flow is more diluted. Flows with high sediment concentration arrive at similar times to the confluence, such that the combination of flows from both streams propagates along the channel keeping a faster wave-front velocity.

## 4.2 Peak flow and its arrival time

By comparing the hydrographs computed using different sediment concentrations in the four points monitored upstream of the confluence, we observe that the most important difference is the magnitude of the peak flow for different concentrations. The relative difference between the peak discharge simulated with clear-water flow compared to a sediment concentration of 60% is 44% at QR-U, and 67% at QR-D. To remove the effects of the additional volume that are produced by the sediment concentration in each stream, in Figures 7 and 8 we show the normalized hydrographs, which are obtained by dividing the discharge by the total volume of the mixture that we obtain at each gauged point defined in Figure 3. We can observe that the difference in the peak flows for different concentrations is only produced by the bulking effect of the sediments.

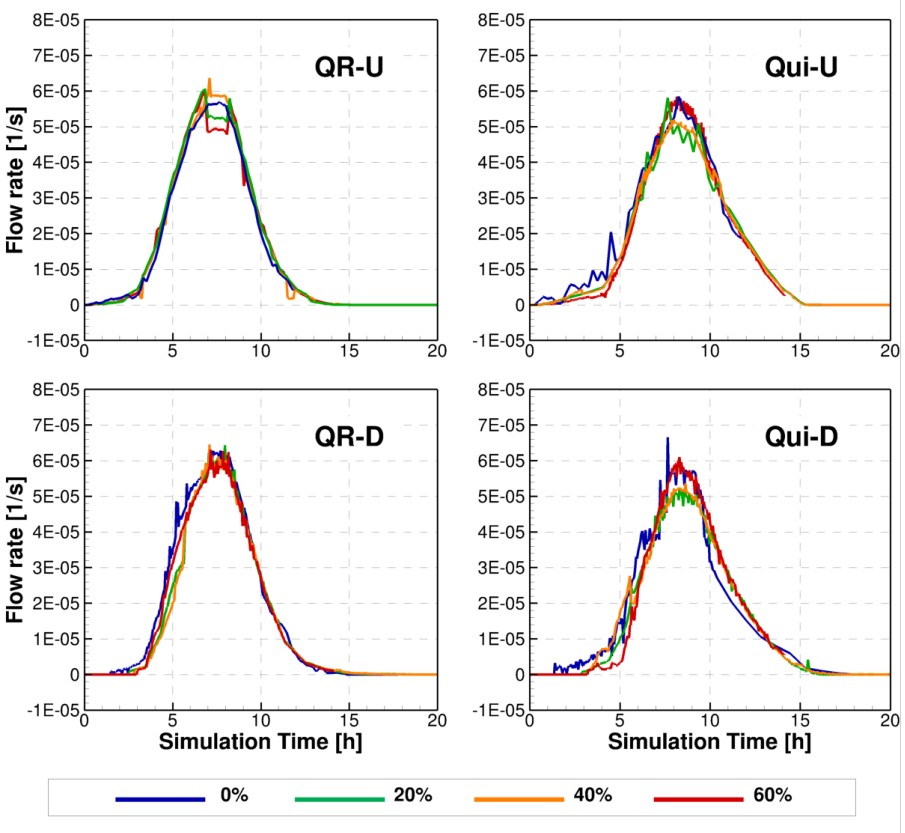

**Figure 7.** Normalized hydrographs at the gauged points upstream of the confluence. In the right panel, we show the upper and lower zones in the Quebrada de Ramón stream, and in the left panel, the points at a similar elevation in the Quillayes stream.

In both streams, the time to the peak of the hydrograph, however, is not significantly affected by the different concentrations. This seems to be related to the shape of the inflow hydrograph, and to the location of the gauged points. The time to reach the peak discharge is around 7 h from the start of the simulation at QR-U, and 10 min later the maximum discharge reaches QR-D. At the Quillayes stream, the peak flow reaches Qui-U after 8 h from the start of the simulation, and Qui-D 13 min later.

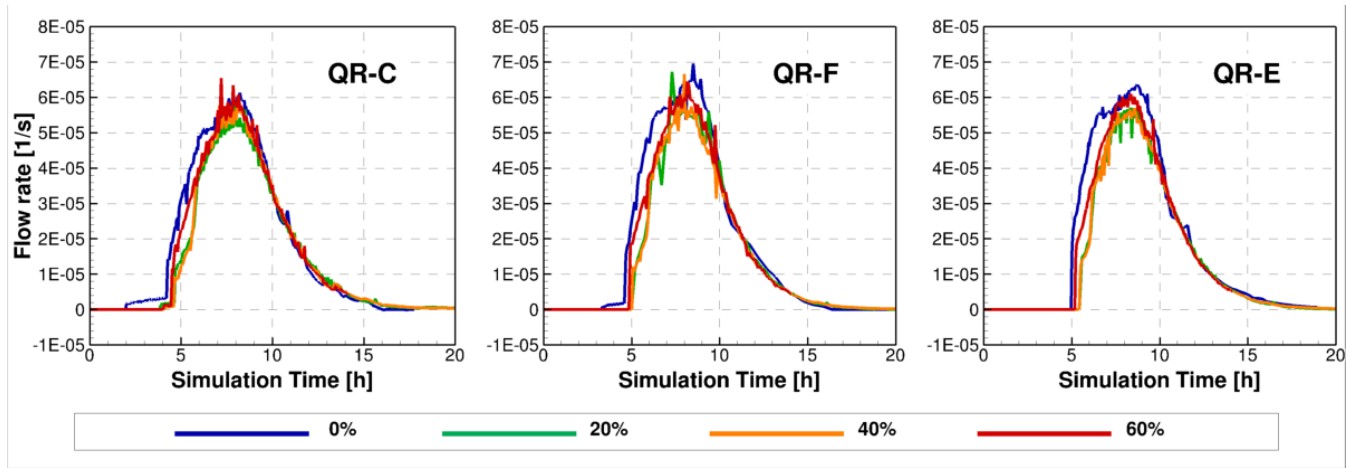

**Figure 8.** Normalized hydrographs at the gauged points downstream of the confluence. In the left panel we depict the point closest to the confluence; in the middle the flooding zone; and in the right panel the outlet of the computational domain.

When we analyze the hydrographs downstream of the confluence we observe similar results, as shown in Figure 8. Due to the progressive reduction of the bed roughness, the peak flows increase in sections closer to the outlet of the watershed. The maximum peak flow is reached at the station QR-F, since the city park located at the north side of the main channel, and between QR-F and QR-E, is flooded and attenuates the peak flow near the outlet.

### 4.3   Total flooded area

The total area in the watershed that is inundated for different sediment concentrations is depicted in Figure 9. No significant differences are noticed for most of the length of the river channel. Both streams have steep slopes in confined canyons, and the maximum flow depth, reaching up to 3 m, do not alter significantly the horizontal extension of the 2D area affected by the flood.

Major differences, however, appear in regions with milder slopes, around the confluence and in the city, near the outlet of
the watershed. At the confluence, simulations with higher concentrations of $40\%$ and $60\%$ overflow the natural channels, due to the fast arrival of a large volume of the water and sediment flow to this region. In the city, near the outlet of the domain, all the flows inundate the urban park located at the north bank of the main channel, downstream of a large urbanized area. In this section, the most important increment of the total flooded area occurs when we increase the concentration from clear water to $20\%$, where the total flooded area increases by $36\%$. For larger concentrations the affected area grows gradually compared to

the clear-water flooding case, as the fluid is more concentrated. Increments of the total area of 46% and 75% are observed for concentrations 40% and 60%, respectively.

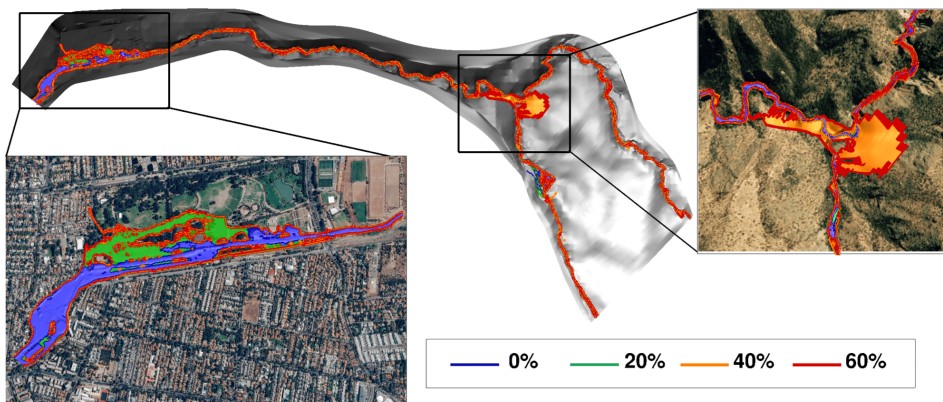

**Figure 9.** Contours of the flooded area for different sediment concentrations along the channel, and in the city of Santiago.

### 4.4 Maximum flow depth and mean velocity

Figure 10 shows the maximum depth registered at each gauged point along the channels for the different sediment concen-
trations we simulate. The numerical results show that the depth increases with concentration, and the largest differences are obtained between the clear-water case and 20% concentration, in 6 of the measurement points we analyze. In QR-C for instance, the first increment of the sediment concentration, from clear-water to 20%, produces a maximum depth that is 24.1% larger, whereas increasing the concentration from 20% to 40%, and then to 60%, the flow depth increases in only 7.5% and 5.1%.
By comparing the flow depths in the simulations, we note that the deepest flow is always located downstream of the confluence (QR-C). In this location, a difference of 0.80 m is measured between the clear-water and the flow with the maximum concentration of 60%. In the urban areas (points QR-F and QR-E), depths larger than 2 m are observed. Here, it is important to have a precise solution for different sediment concentrations, since there is a difference of 0.5 m between clear-water and the maximum concentration, which can have significant impacts on the design of flood control measures.
Additionally, in Figure 11 we show the mean velocity at each measurement point for the range of sediment concentrations. In this case, we cannot observe a clear trend of velocity changes as a function of concentration. For this flow variable, it seems that the local topographic conditions affect considerably the averages of the flow hydrodynamics.

In Figure 12 we relate the magnitude of the hydrodynamic variables, velocity vs depth, computed at each time step at QR-U on the right panel, and Qui-U on the left panel. These plots are similar to a stage-velocity relation that links the flow depth and
the total velocity at the same instant in time. The plots are constructed using data every 30 s, for a total time of 24 hours.

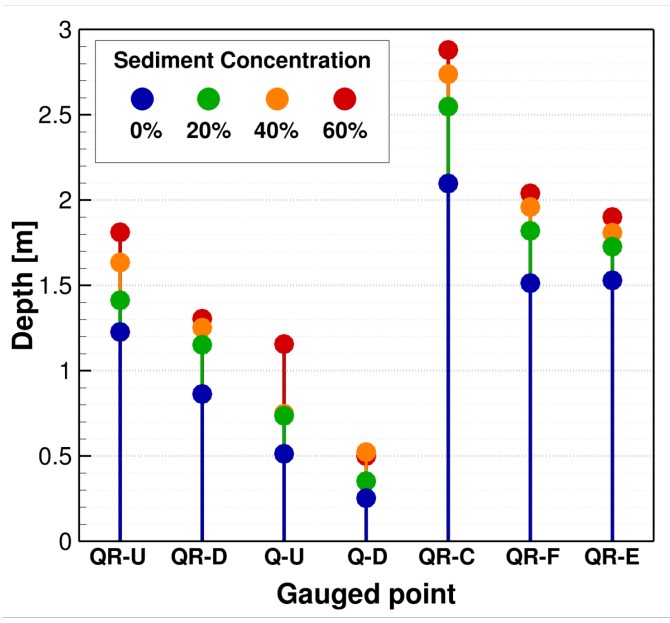

**Figure 10.** Maximum depth computed in every gauged point depending of the sediment concentration.

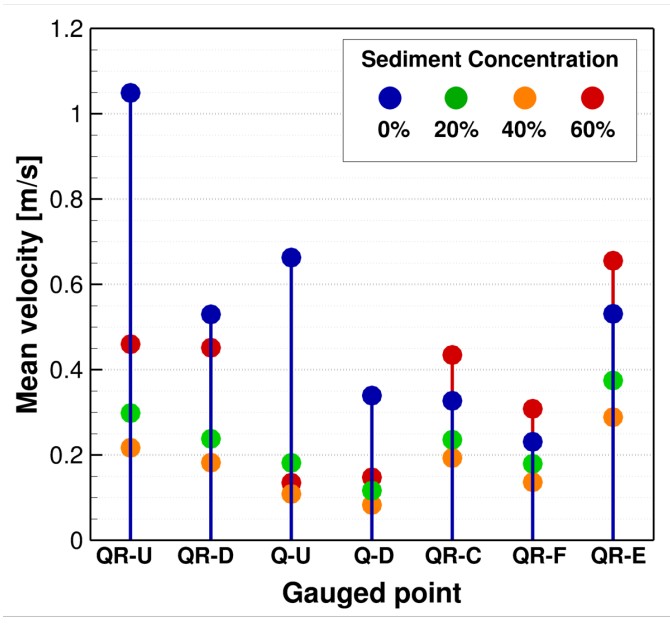

**Figure 11.** Mean velocity computed in each gauged point depending on the sediment concentration.

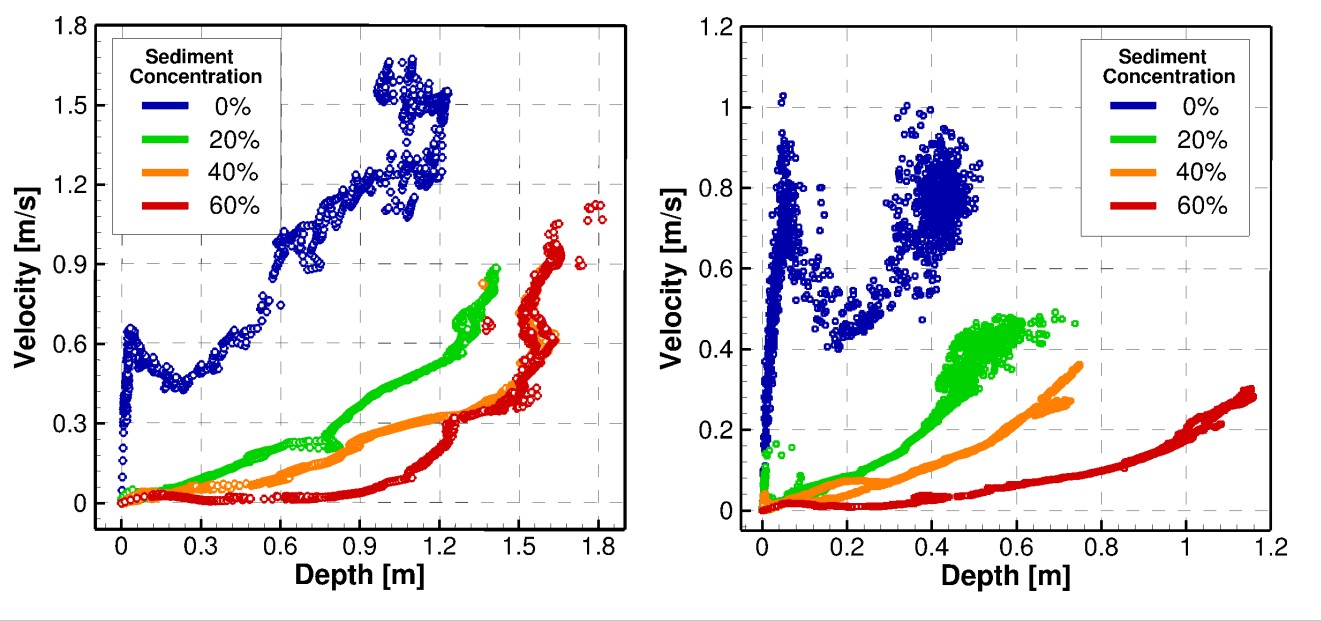

**Figure 12.** Stage-velocity relation between the computed flow depths and velocities for different sediment concentrations. The data at the point QR-U is shown on the left, and Qui-U on the right.

Even though a direct relation is not observed between mean velocity and sediment concentration in points QR-U and Qui-U (Figure 11), the depth-velocity plots in Figure 12 confirm the relation of large depths and lower velocities as the concentration increases. This is closely related to the changes on the flow resistance. Larger concentrations increase the yield stress, the fluid viscosity, and the dispersive effects, producing additional momentum losses, which reduce the flow velocity. The stage-velocity
relation is different for clear-water flow as compared to the sediment laden cases. For hyperconcentrated flows, the relationship between the depth and velocity is fitted to a quadratic regression that always increases. In clear water the relation is linear in shallow flows under 0.05 m deep, where the Froude number is larger than 1, reaching 1.43 and 1.53 in the QR-U and Qui-U respectively. Then, a transition zone with depths between 0.05 m and 0.2 m, and almost critical Froude number is observed. Above 0.2 m depth, the velocity increases quadratically as seen in the flows with higher sediment concentrations. In this zone,
the flow is dominated by gravity with subcritical Froude numbers of around 0.25.

### 4.5    Flow momentum in the urban area

To evaluate the potential damage to the infrastructure generated by floods, we can compute the flow momentum at each cross-section of the flooded area. In this case we compare the maximum force produced by the flow in the urban area of the watershed, considering flows with different sediment concentrations coming from the Quebrada de Ramón and the Quillayes
stream. Figure 13 shows contours of maximum cross-section momentum along the river. The top figure shows the momentum

for a sediment concentration of 20% in the Quebrada de Ramón and 60% in the Quillayes stream. The opposite case, 60% in the Quebrada de Ramón and 20% in the Quillayes stream, is depicted in the lower image.

The approaching flow has an approximate force of 700 kN in both simulations. For these two cases, the areas with highest momentum correspond to: (1) The confined zone on the right of the image; and (2) At the outlet of the basin in the urban area. However, the force is on average 14.5% higher in the second case, which could be related to the higher flow density of the flow that is obtained downstream of the confluence.

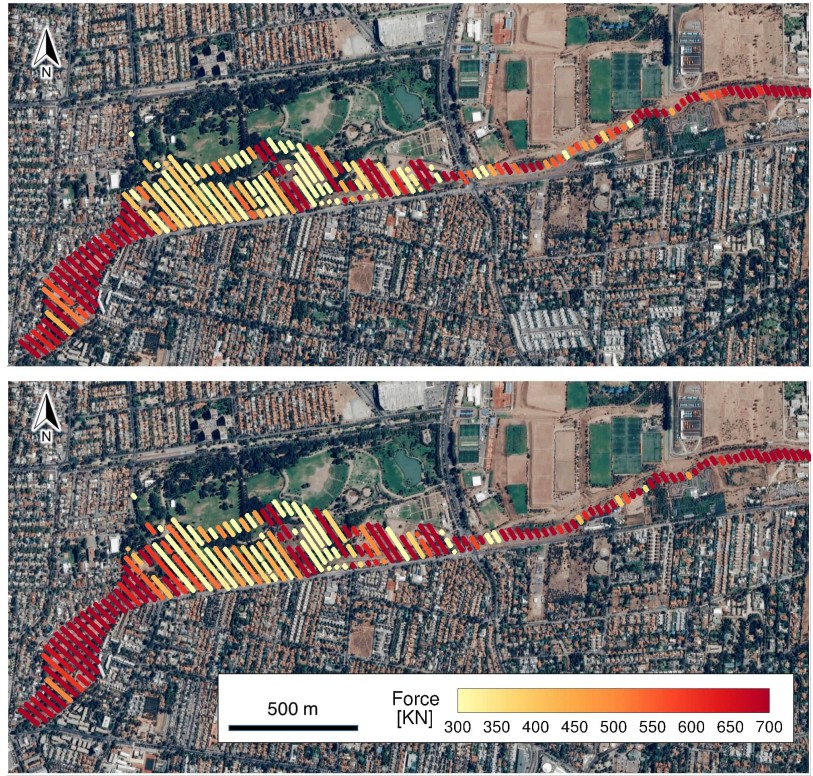

**Figure 13.** Maximum momentum of the flow in the urban area. The picture above shows results for a concentration of 20% coming from the Quebrada de Ramón and 60% from the Quillayes stream. The image below corresponds to the opposite case, 60% from the Quebrada de Ramón and 20% from the Quillayes stream.

Since the density in these simulations is different in both streams upstream the confluence, the density of the fluid in the main channel varies in time and space, both along and across the flow. The mean concentration in the main channel, downstream of the confluence, is around 30% and 44% considering a sediment concentration of 20% in the Quebrada de Ramón and 60% in the Quillayes stream and viceversa, respectively. These values are consistent with the theoretical concentrations computed from the fully mixed conditions.

## 5   Discussion

The results evidence the competition between two main factors that control the dynamics of the flow in mountain rivers at different spatial and temporal scales: (1) The geomorphological features of the river represented by the bathymetry, the slope and the channel width; and (2) The flow resistance due to the internal sediment dynamics that changes the rheology of the mixture.

At time scales of seconds or minutes, flow velocities and depths along the channel are significantly affected by both factors, having a great impact on global variables such as the wave-front velocity, the total inundated area, and the cross-section momentum of the flow. As reported in section 4.1, the mean velocity of the wave front is reduced as the sediment concentration increases, which is produced by flow resistance driven by the dispersive stresses of the sediment. The flow depth increases by these higher momentum losses, and by the bulking effect of the additional sediment mass, resulting on a direct impact on the increment of the flooded area and flow momentum in the city. We also observe discontinuities on the advance of the flow front in time, which are located in the vicinity of sudden changes of slope or channel width that are common in mountain canyons in the Andes. Local changes of sediment concentration can even suppress geomorphic effects, having large-scale impacts on the flood propagation, as the sediment concentration can change the flow regime from supercritical to subcritical, as shown in Fig. 12.

Global bulk variables, on the other hand, such as the normalized hydrograph shape and the time to the peak discharge, show a geomorphic control at the scales of the duration of the entire event. Except on areas where there is a change on the flow regime, the effects of the sediment concentration are not observed for the time-averaged velocities along the channel. As shown the normalized plots in Fig 7, however, the differences on the arrival time of the peak flow of the order of minutes, which is small compared to the entire duration of the flood hydrograph with a total of 20 hours.

It is important to point out that the sensitivity of the flow physics affected by the sediment concentration, such as the mean velocity of the wave front, flow depth, instantaneous velocity, flooded area and flow momentum, decreases for higher sediment concentrations. We show that as the sediment concentration increases, the changes are more significant in the range between 0–20%, compared to the flood propagation for increments over 40%. These new insights are relevant to determine flood hazard in mountain rivers, and define a reduced number of possible scenarios for different concentrations in these rivers.

## 6   Conclusions

The primary emphasis of this work is to examine the effects of the sediment concentration on the flood dynamics in an Andean watershed. To simulate different scenarios, we developed a finite-volume numerical model that solves the hydrodynamics of hyperconcentrated fluids in complex natural topographies. The model is based on the work of Guerra et al. (2014), and it is employed to solve the non-linear shallow water equations coupled to a transport equation for the sediment in generalized curvilinear coordinates. To consider the effects of the sediment concentration, we implement a new version of the quadratic rheological model (O'Brien et al., 1993) to calculate the stresses produced by high concentrations, separating the turbulent and dispersive effects of the sediment concentration.

To investigate the effects of the sediment concentration in floods that occur in mountain rivers, we perform simulations in the Quebrada de Ramón watershed, an Andean catchment located in central Chile. We analyze the changes on hydrodynamic variables such as peak discharge, arrival time of the flood wave, cross-section momentum, flow depth, mean velocity, and total flooded area. Most the these results are compared and analyzed in seven points along the channel.

The most important effects on the flood propagation are observed for the increments of sediment concentration just above the clear-water flow, in the range of concentrations from $0\%$ to $20\%$. Even though the channel slope is the most important morphological feature that controls the dynamics of the flow, local factors such as channel widening can change significantly the propagation of the flood wave. High sediment concentrations modulate these morphodynamic effects, producing larger flow depths and slower velocities overall.

Some of the hydrodynamic variables analyzed were more sensitive to changes in sediment concentration. We observed significant effects on the total flooded area and momentum of the flow as the flood arrives to the urban area. While the extent of the 2D flooded area in the entire basin remains more or less constant for different concentrations, the largest difference is observed in the city, where the slopes are milder. The simulations show a difference of $76\%$ in the total 2D flooded area when we compare the clear-water conditions with the $60\%$ concentration. Regarding the cross-section momentum as the flood advances in the urban zone, we show that the maximum momentum of the flow increases $14\%$ on average for a $20\%$ concentration in the Quebrada de Ramón, and $60\%$ concentration in the Quillayes stream. We also observe that bulk variables, such as the arrival time of the peak discharge at different locations of the basin, and the shape of the hydrograph, are not modified significantly with the magnitude of the sediment concentration, but they are associated to the local morphological conditions of the river channel.

## Appendix A: Numerical method

The numerical solution of the system of equations 5 is based on the method developed by Guerra et al. (2014) to solve the NSWE, which has shown great efficiency and precision to simulate extreme flows and rapid flooding over natural terrains and complex geometries. This is a finite-volume formulation that is implemented in two steps: First, in the so-called hyperbolic step, the Riemann problem is solved at each element of the discretization without considering momentum sinks. The flow is reconstructed hydrostatically from the bed slope source-term, adding the effects of the spatial concentration gradients. In the second step we incorporate the shear stress source terms by means of a semi-implicit scheme, correcting the predicted values of the hydrodynamic variables obtained in the previous step.

The initial hyperbolic step consists of solving numerically the following equation:

$$\frac{\partial Q}{\partial t} + J\frac{\partial F}{\partial \xi} + J\frac{\partial G}{\partial \eta} = S_B(Q) + S_C(Q) \tag{A1}$$

in which a semi-discrete finite-volume formulation in generalized curvilinear coordinates can be written as follows,

$$\frac{\partial Q_{i,j}}{\partial t} + \frac{J_{i,j}}{\Delta \xi}\left(F_{i+\frac{1}{2},j} - F_{i-\frac{1}{2},j}\right) + \frac{J_{i,j}}{\Delta \eta}\left(G_{i,j+\frac{1}{2}} - G_{i,j-\frac{1}{2}}\right) = S_{B_{i,j}} + S_{C_{i,j}} \tag{A2}$$

where $Q_{i,j}$ and $J_{i,j}$ represent the vector of hydrodynamic variables and the Jacobian of the coordinate transformation at the center of the discrete elements of the grid $(i,j)$. The vectors $F_{i\pm\frac{1}{2},j}$ and $G_{i,j\pm\frac{1}{2}}$ are the numerical fluxes through each of the cell interfaces. The terms $\Delta\xi$ and $\Delta\eta$ correspond to the size of the discretization, and $S_{B_{i,j}}$ and $S_{C_{i,j}}$ are the discrete source terms of the bed slope and concentration gradients, respectively.

To compute the numerical fluxes we implement the VFRoe-ncv method (Gallouët et al., 2003; Marche, 2006) to solve equation A1, through a non-conservative change of variables, linearizing the Riemann problem (Guerra et al., 2014). The vector of hydrodynamic variables $Q_{i,j}$ is extrapolated to the boundaries of the each cell to ensure the non-negativity of the intermediate states and flow depths, preserving the dry zones of the terrain. The Monotonic Upwind Scheme for Conservation Laws method (MUSCL), developed by Van Leer (1979), is used to perform the extrapolation with second order accuracy.

Finally, the methology developed by Masella et al. (1999) is used to avoid unphysical solutions due to the lack of dissipation to capture shock waves.

     The bed-slope source term $S_{B_{i,j}}$ is computed following the well-balanced methodology developed by Audusse et al. (2004) and adapted to generalized curvilinear coordinates by Guerra et al. (2014). This method reconstructs hydrostatically the free surface by performing a balance between the topographic variations of the domain and the hydrostatic pressure. The hydro-

dynamic variables and bed elevations are extrapolated to the boundaries of the cells using the MUSCL method, preserving locally and globally the dry zones and stationary steady-states. To ensure the non-negativity of the flow depth and to avoid spurious oscillations, the *minmod* limiter is implemented during the hydrostatic reconstruction of the fluid depth; such that realistic values of the spatial gradients of depth are reached in the shock waves (LeVeque, 2002; Bohorquez and Fernandez-Feria, 2008).

     The concentration gradient term $S_{C_{i,j}}$ is discretized using the following scheme:

$$S_{C_{i,j}} = \frac{-h_{i,j}^2}{2Fr^2}\left(\frac{s-1}{C_{i,j}(s-1)+1}\right)\left(\frac{C_{i+\frac{1}{2},j}-C_{i-\frac{1}{2},j}}{\Delta\xi}\xi_x + \frac{C_{i,j+\frac{1}{2}}-C_{i,j-\frac{1}{2}}}{\Delta\eta}\eta_x\right) \tag{A3}$$

where, $h_{i,j}$ and $C_{i,j}$ are the centered-cell flow depth and sediment concentration, respectively, $C_{i\pm\frac{1}{2},j}$ and $C_{i,j\pm\frac{1}{2}}$ are the sediment concentrations at the interfaces of the each cell, obtained from a first order upwind scheme.

     In the second step of the numerical solution, we incorporate the momentum source terms in vector $S_S(Q)$, solving the following system of equations,

$$\frac{\partial hu}{\partial t} = -S_x; \quad \frac{\partial hv}{\partial t} = -S_y \tag{A4}$$

We use a splitting semi-implicit method (Liang and Marche, 2009), employing a second order Taylor expansion. The limiters developed by Burguete et al. (2007) are implemented to avoid numerical instabilities at the wet/dry interfaces, where the flow depths are shallower. These limiters are designed to prevent unphysical effects, such as reversed flows due to high shear stresses.

Finally, the temporal integration of equation A1 is carried out by using a fourth-order Runge-Kutta numerical scheme. The condition for numerical stability of the model is based on the Courant-Friedrichs-Lewy criterion $CFL$.

     The boundary conditions are handled by creating two rows of "ghost-cells" outside of the computational domain (Sanders, 2002). We implement three types of boundaries: (1) Open or transmissive boundary at the outlets; (2) Closed reflective bound-

ary for the solid walls; and (3) Inflow boundary to introduce a hydrograph or a controlled discharge toward the computational domain.

## Appendix B:  Tests for the density coupling

### B1    Quiescent equilibrium in a tank

This benchmark test is developed to demonstrate the capacity of the model to preserve the hydrostatic state with density differences. An analytical solution is obtained from the procedure developed by Leighton et al. (2009), in which the original system of equations 1 to 4 is simplified by considering steady flow ($\frac{\partial(\cdot)}{\partial\hat{t}} = 0$) and a stationary initial state ($\hat{u} = \hat{v} = 0$) in inviscid flow with zero stresses ($\hat{\tau}_{\hat{x}} = \hat{\tau}_{\hat{y}} = 0$). Therefore, the equations are reduced to:

$$\frac{\partial}{\partial x}\left(\frac{1}{2}\hat{\rho}g\hat{h}^2\right) = -\hat{\rho}g\hat{h}\frac{\partial\hat{z}}{\partial\hat{x}} \tag{B1}$$

which can be written as follows:

$$\frac{\hat{h}}{\hat{\rho}}\frac{\partial\hat{\rho}}{\partial\hat{x}} + 2\frac{\partial\hat{h}}{\partial\hat{x}} = -2\frac{\partial\hat{z}}{\partial\hat{x}} \tag{B2}$$

Hence, for a rectangular tank of length $L$ and width $A$, with a constant initial flow depth $\hat{h}(\hat{x}) = h_0$, and a bed described by a cosine function:

$$\hat{z}(\hat{x}) = A\left[1 - \cos\left(\frac{2\pi\hat{x}}{L}\right)\right] \tag{B3}$$

the analytical solution of equation B2 becomes:

$$\hat{\rho}(\hat{x}) = \rho_0 \exp\left[\frac{2A}{h_0}\cos\left(\frac{2\pi\hat{x}}{L}\right)\right] \tag{B4}$$

where $\rho_0$ is the initial reference value of the fluid density.

The dimensions and initial conditions of this test are presented in the Figure B1, where the reference density $\rho_0$ was set to 1,000 kg/m$^3$. The 1D computational domain was discretized in a grid of $1,001$ cells on the longitudinal direction, with a
reflective solid wall boundary condition at each wall of the tank. The total simulated time was $100$ s and the CFL number was set to $0.2$.

Results show that there is an excellent agreement between the analytical and numerical solution for the free surface, as shown in Fig. B2(a), and the density profile in Fig. B2(b). The maximum error of water depth is equal to $10^{-7}$ m, and the model is capable of maintaining the steady-state of the flow.

### B2    Density dam-break with two initial discontinuities

To test the model in unsteady conditions, we simulate a density-driven dam-break to evaluate the evolution of the hydrodynamic variables in space and time. The numerical experiment is based in the work developed by Leighton et al. (2009), which consists

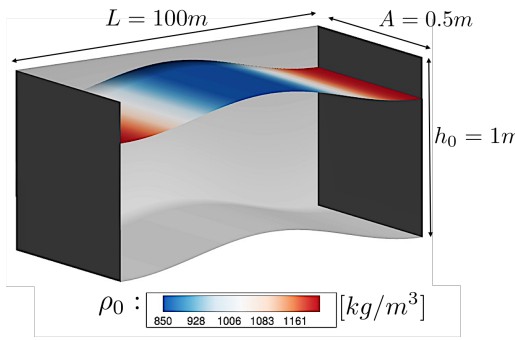

**Figure B1.** Dimensions and initial conditions of the rectangular tank used for the quiescent equilibrium test (Leighton et al., 2009).

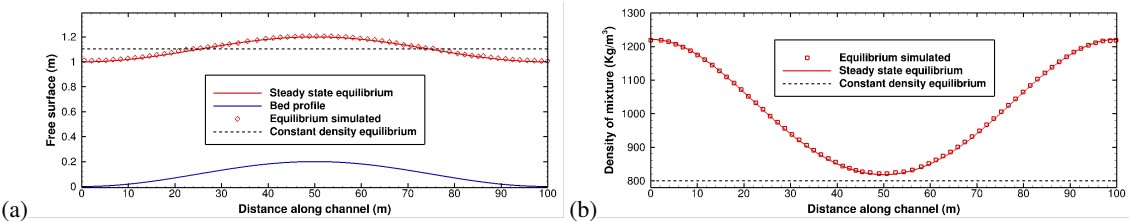

**Figure B2.** Quiescent equilibrium test. Comparison between theoretical and numerical profiles of hydrodynamic variables. (a) Free surface; and (b) Fluid density.

of a horizontal rectangular tank of 100 m long, with two fluids of different densities $\rho_1$ and $\rho_2$, as shown in Figure B3. The acceleration of gravity is considered equal to 1 m/s$^2$, and the shear stresses are neglected.

Two different simulations are performed for $\rho_2 = 0.1$ kg/m$^3$ and $\rho_2 = 10$ kg/m$^3$, while $\rho_1$ is kept constant and equal to 1 kg/m$^3$. Both simulations are implemented on a regular grid with a resolution of 0.005 m during 30 s, using a $CFL = 0.2$ and reflective boundary conditions at solid walls.

In Figure B4 we show the instantaneous flow depth and velocity profiles at 2 and 30 s, from the start of the first simulation ($\rho_2 = 0.1$ kg/m$^3$). A good agreement is found with respect to the solution provided by Leighton et al. (2009), as we capture the propagation of the free surface and velocity magnitudes that are generated by the initial imbalance of the hydrostatic pressure at the interface of the fluids. The amplitudes of the main shock are slightly smaller due to the second-order accuracy of the numerical model. Similar results are obtained for $\rho_2 = 10$ kg/m$^3$ as shown in Figures B5 and B6, which also shows that the model can resolves sharp gradients, and the solutions do not change significantly with the grid resolution.

Note that when $\rho_2/\rho_1 < 1$, the flow velocities are directed toward the middle of the tank, where the fluid is less dense, which increases the flow depth in that zone. Conversely, when $\rho_2/\rho_1 > 1$, the fluids move to reach hydrostatic equilibrium, balancing the pressure in the entire domain, which produces higher depths at the sidewalls.

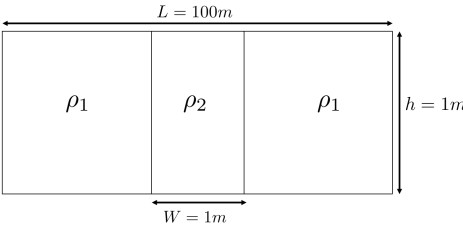

**Figure B3.** Initial state of the density-driven dam-break (Leighton et al., 2009).

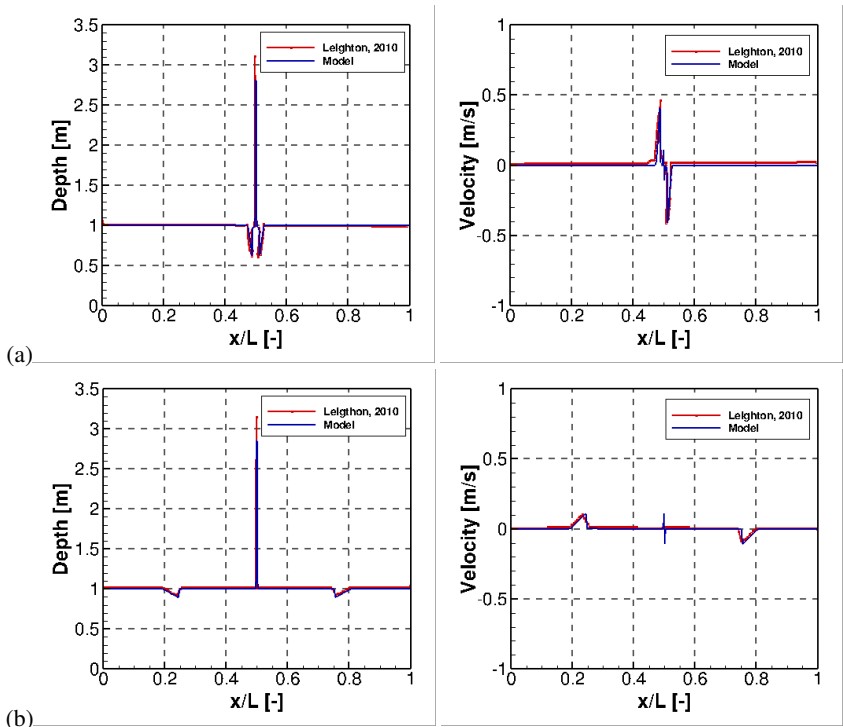

**Figure B4.** Density driven dam break. Case $\rho_2 = 0.1$ kg/m$^3$: Comparison of the flow depth (on the left) and velocity profiles (on the right) at (a) $\hat{t} = 2$ s; and (b) $\hat{t} = 30$ s from the beginning of the simulation.

## B3  Large-scale experimental dam-break

To test the rheological model, we simulate the large-scale dam-break experiment with high sediment concentration performed by Iverson et al. (2010). We compare the numerical results with the measurements of flow depth and the arrival time of the wave front. It is important to note that the simulation of this experiment is a very challenging computational test for the numerical
5    model. The slope of the channel, the sediment concentration, and the flow phenomena as the wave advances generates a complex dynamic that is difficult to measure and reproduce with a high resolution numerical model.

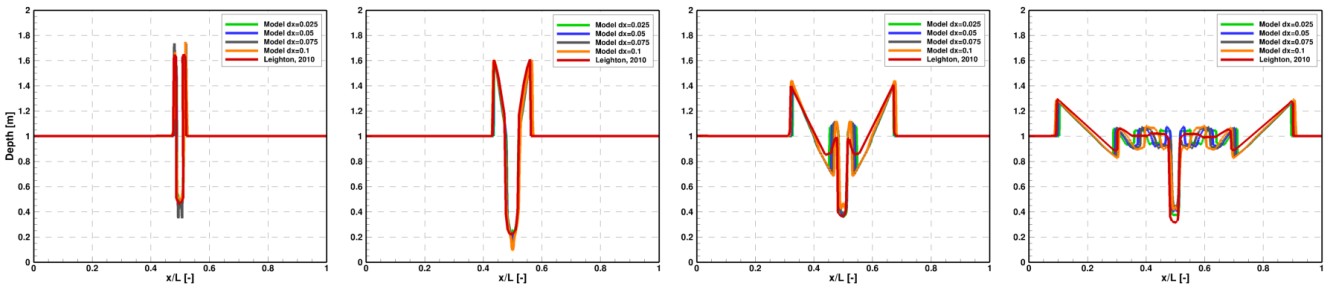

**Figure B5.** Density driven dam break. Case $\rho_2 = 10$ kg/m$^3$: Comparison of the flow depth at $\hat{t} = 1$ s, 4 s, 12 s, and 30 s from the beginning of the simulation.

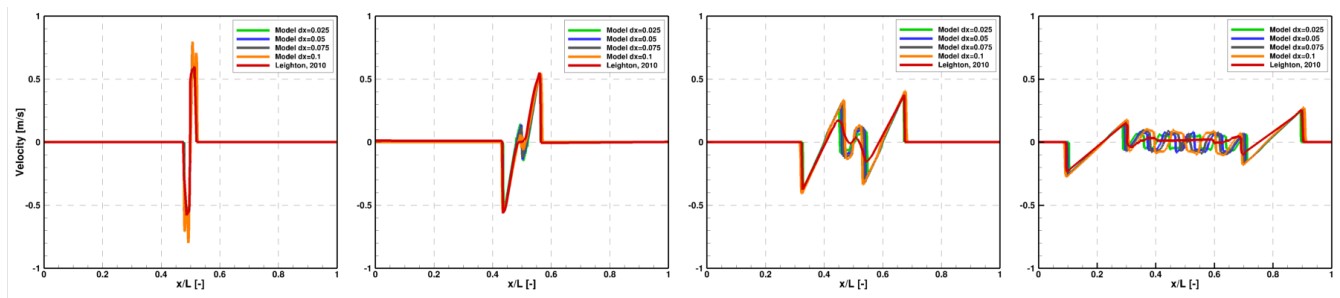

**Figure B6.** Density driven dam break. Case $\rho_2 = 10$ kg/m$^3$: Comparison of velocity profiles at $\hat{t} = 1$ s, 4 s, 12 s, and 30 s from the beginning of the simulation.

The experiment consists of the sudden release of a large volume of a sediment-water mixture on a 95 m long rectangular channel, with a cross section of 2 m wide by 1.2 m deep. The channel is very steep, with an inclination of $31°$ on the first 75 m downstream from the gate, and $2.5°$ on the downstream section. The total volume released in the dam-break experiment is 6 m$^3$, with an initial depth of 2 m and a volumetric sediment concentration of $64.7\%$. The bed roughness changes along the channel, with a representative roughness height of 1 mm on the first 6 m of the channel measured from the gate, and a roughness of 15 mm in the rest of the channel, downstream. The sediment density considered in this case is $\rho_s = 2,700$ kg/m$^3$.

The unsteady inflow condition is the debris flow at a distance of 2 m downstream from the gate, which is shown in Figure B7. This was obtained from the simulation of the dam break delayed 1 s, to consider the delay on the opening of the gate, as reported by Iverson et al. (2010). We simulate a total time of 25 s, using a 2D spatial discretization with a uniform resolution of 0.1 m, and a CFL number equal to 0.1.

In Figure B8 we compare the flow thickness between the simulation and the experiment at two locations, corresponding to 32 and 66 m downstream of the gate. The experimental data was collected by Iverson et al. (2010) at a frequency of 100 Hz. In our simulation, the grid is fine enough to resolve the well-known roll waves that appear at high Froude numbers in steep channels (Bohorquez and Fernandez-Feria, 2006). This phenomenon has also been recently observed in the simulations of the

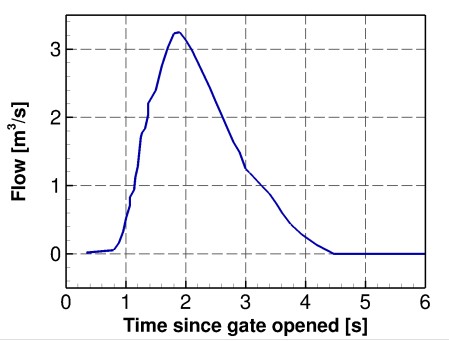

**Figure B7.** Unsteady inflow boundary condition corresponds to the cross-section flow measured at a location of 2 m downstream of the gate (Iverson et al., 2010).

same experiment by Bohorquez (2011). In this case we capture roll waves with an amplitude close to 0.5 m, as shown by the red line in Figure B8.

To compare the numerical results directly with data provided by the experiments, we apply the same moving-average filter used to smooth the experimental data (black line in Figure B8). The model reproduces with good agreement the arrival time of the wave front in both gauges, with delays smaller than 0.2 s. The maximum flow depth computed at the location of 32 m and 66 m downstream from the gate is over- and underestimated by just 1.78 cm and 1.6 cm respectively.

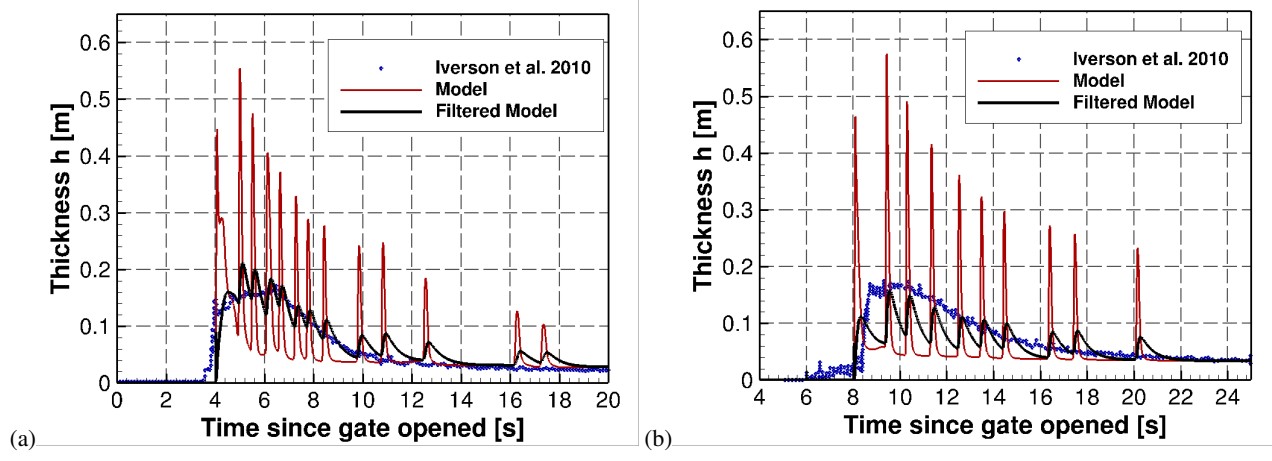

**Figure B8.** Comparison of the flow thickness measured in the experiment of Iverson et al. (2010), and computed with our model. Flow thickness at two locations: (a) 32 m; and (b) 66 m downslope from the gate.

The simulated and observed wave-front position in time are very similar (Figure B9) with values of the mean square error and the coefficient of determination of the fit being 1.98 m and $R^2 = 99.28\%$, respectively. The sudden discontinuity in the

computed front velocity at 7.6 s is due to a roll wave advancing through the front, which briefly slows down the flow. Due to the resolution of the experimental results, we cannot compare directly this phenomenon captured in the simulation.

Overall, the validation study shows that the numerical model in these extreme cases is very robust and it is able to reproduce many of the phenomena of interest that appear in hyperconcentrated flash floods.

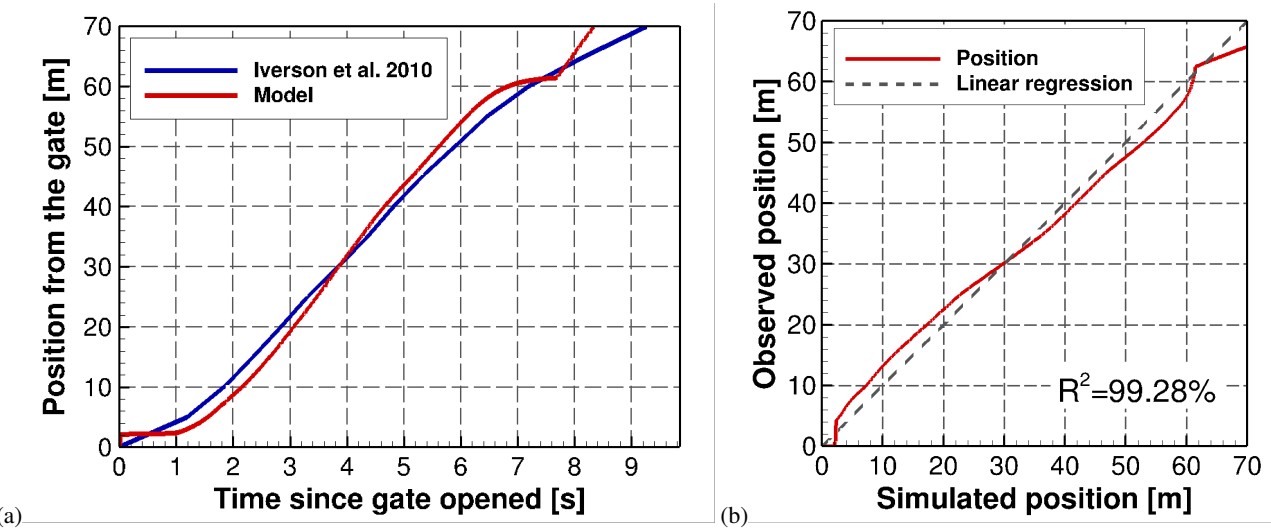

**Figure B9.** Comparison of the position of the flow front as a function of time: (a) The position of the flow front over the time (b) Comparison of the experimental and simulated flow front position. The dashed line is the perfect fit with slope equal to 1.

5  *Competing interests.*  No competing interests are present

*Acknowledgements.*  This work has been supported by Conicyt/Fondap grant 15110017. This research has been partially supported by the supercomputing infrastructure of the NLHPC (ECM-02). Additional funding from VRI of the Pontificia Universidad Católica de Chile, internationalization of research, project PUC1566, MINEDUC. We thank the Ministry of Public Works for providing the LIDAR topography. Ms. Verónica Ríos and professor Jorge Gironás provided the flood hydrographs.

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
