# Peer review of "Modeling the effects of sediment concentration on the propagation of flash floods in an Andean watershed"

_Natural Hazards and Earth System Sciences, 2019_

## Referee Comment (RC1) · Anonymous Referee #1 · 16 Jul 2019

This review is of "Modeling the effects of sediment concentration on the propagation of flash floods in and Andean watershed" by Contreras and Escauriaza. Overall I recommend its publication after minor revisions to clarify a few things and answer some questions.

Abstract: I think it undersells the results, in particular the effects that flash flood sediment concentrations do have. When I read the abstract before reading the rest of the manuscript, my take-home understanding was that the authors found that sediment concentration doesn't really matter (lines 13-15). Sell the results better! An abstract should basically be advertising to get people to read the rest of the paper. The findings of how sediment concentration influence flow depth and the extent of flooding are pretty different from this. Emphasizing how sediment concentration DOES change flooding will get more people to read the manuscript, and is totally legitimate based on the results. Also, I think there should be more results in the abstract; if length is an issue, then shorten other parts of the abstract. For example, the introductory material in lines 1-6 is fine, but could be shortened to 1-2 lines if needed. Right now, really the only results are lines 12-15. Finally, I think add more numbers to the abstract, such as percent changes in flood depth or flood extent resulting from changes in sediment concentration.

Pg3 line 24 to pg4 line 4: shorten this; it feels repetitive. Its basically two out-lines/roadmaps of the paper back to back.

I confess that I did not check the equations in great detail for errors; apologies.

Pg 5 line 23: suggest changing to "...we do not consider erosion or deposition of the bed."

Pg 5 line 6: Unclear to me—is the model validation what you show in Appendix A, or is it previous work that needs to be cited, like Guerra et al. 2014? I think the answer is Appendix A; I suggest rewording a bit to make it clear that you present the validation in Appendix A.

Pg 5 line 11: here and elsewhere, suggest cutting "for details" from citations. Its not needed.

Pg 7 line 26: this is my ignorance, but I don't know the difference between turbulent and dispersive stresses. It would be helpful to readers like me to work in a 1 or 2 sentence explanation. I see that you address this a bit on the next page (and cite Julien and Paris 2010), but I still suggest a little more.

Pg 8 line 20: suggest changing "sediment concentrations" to "volume sediment con-centrations"

Figure 1: I suggest somehow indicating which channel here is the mainstem and which is the Quillayes tributary. I can guess or probably figure it out from figure 3, but would be helpful to have more obvious on this figure. Also, in the caption its unclear to me which black line you're talking about—looks to me like there's a thin black line around the gridded part, and a thicker black line around the entire watershed.

Figure 2: while I believe the figure shows a confluence, I can't actually figure out where another stream comes in. Maybe annotate on the photo where the other channel is? Or use a different picture showing it more clearly? I do like the action shows with at least one of the authors for scale.

Figure 4: suggest changing y axis to Discharge rather than Flow

Pg 11 line 8-9: If CFL was defined earlier I missed it; make sure to define it.

Table 4.1: Some percentages are given in the text, but I suggest just adding columns of "% change from clear water" or similar to the table. I'm surprised at how much difference sed conc makes and think that showing percentages (and editing the abstract) would emphasize this more.

Pg 12 line 10: I'm hesitant to say its an exponential decrease unless there's a plot or other curve fitting showing that an exponential really does work well. Not everything that changes magnitude with time follows exponential decay.

Pg 13 line 3 (at least as the line #s show up on my pdf; its actually farther down the page): I disagree that figure 6 shows "similar dynamics" to figure 5. I'm confused by this. Figure 6 is practically inverted from Figure 5. In figure 6, the clear water case propagates slowest, 60% propagates fastest (if I'm not confused), which is opposite. There's a bit of possible explanation at the bottom of the page (it has something to do with different arrival times of flood waves on tributaries?), but I don't really understand. I think change "similar dynamics" and explain more what causes the differences between these figures.

Pg 15 line 16 (I think line #s are messed up on the version I'm reading; it's a couple lines above figure 8): change "around of 7 h" to "around 7 h"

Pg 15 line 25: change "maximum increments of flow depth" to "maximum flow depth"? I may be misinterpreting, but I don't think you're talking about 3 m depth increments, I think you're talking about total flood depths at a given location of ~3m.

Figure 10 and 11: Change Q_U to Qui_U, as is used elsewhere in manuscript.

Pg 20 line 26: change sensibility to sensitivity?

Pg 21 line 20: change extension to extent.

Figure B1 caption: change "used to the quiescent..." to "used for the quiescent..."

Figure B3: Unclear to me where the dam is located. I suggest adding more explanation to the caption, to explain that the "dams" are between rho1 and rho2? Also suggest in the caption saying that h is the flume width; I was confused a bit about h vs w.

---

## Referee Comment (RC2) · Anonymous Referee #2 · 19 Sep 2019

Review of:

**Modeling the effects of sediment concentration on the propagation of flash floods in an Andean watershed**

By
**María Teresa Contreras1,2,3and Cristián Escauriaza1,2**

This article presents an analysis of flashflood propagation in a steep watershed in the Andes. The article should be of interest to the readers of this journal. The following should be considered for improving this article:

1. The objectives are not quite clear. We all understand the emphasis on the model and the effects of sediment concentration, but what is it that you are particularly wanting the reader to learn from your research? The model and color figures are nice, but there must be some scientific objective that you tried to accomplish and want to share with the readers.
2. The conclusions are long and a bit vague. There should be a clear delineation of what can be concluded from this analysis. Also, the wishful thinking at the end of what you want to do in the future should be left out. There should be a greater emphasis on what has been done and what can clearly be demonstrated from your analysis. What can be achieved in the future should be left out for your next paper…
3. The article is a bit long and there is quite a bit of excess verbiage (a good 10-15% can be trimmed out) that could be deleted without changing the technical content of your discussion. Also, once the paper is approved for publication, it seems better not to include the Appendix in this paper. This material can be useful to the reviewers at this stage of the review process, but will not be necessary in the final paper.
4. The analysis of the effects of sediment concentration is interesting, but the results at a 60% concentration seem too fluid and flowing quite fast. Depending on the amount of clay and the type of clays, the flows at such a concentration can be very different than modeled. These hyper-concentrated flows may also resembling very slow moving mud flows. It may also be useful to indicate whether this is a concentration by weight or by volume. It does make a large difference at high concentrations.

Overall, this article is very interesting and should be published once minor changes and improvements are carried out. I can re-review a modified version if needed, but the modifications suggested above may simply be implemented to the satisfaction of the Editorial Board.

---

## Author Comment (AC1) · 15 Nov 2019

*Response to Referee #1*

We wish to thank the Reviewer for his/her thorough review of our manuscript and for the useful comments that helped us improve the quality of the paper. The specific issues raised by this Referee are addressed in detail below:

*Abstract: I think it undersells the results, in particular the effects that flash flood sediment concentrations do have. When I read the abstract before reading the rest of the manuscript, my take-home understanding was that the authors found that sediment concentration doesn't really matter (lines 13-15). Sell the results better! An abstract should basically be advertising to get people to read the rest of the paper. The findings of how sediment concentration influence flow depth and the extent of flooding are pretty different from this. Emphasizing how sediment concentration DOES change flooding will get more people to read the manuscript, and is totally legitimate based on the results. Also, I think there should be more results in the abstract; if length is an issue, then shorten other parts of the abstract. For example, the introductory material in lines 1-6 is fine, but could be shortened to 1-2 lines if needed. Right now, really the only results are lines 12-15. Finally, I think add more numbers to the abstract, such as percent changes in flood depth or flood extent resulting from changes in sediment*
*concentration.*
**Response:**

Thank you for your kind assessment of our work. We have modified the abstract by reducing the introduction, clarifying the objective of our research, and summarizing all the main results of our work. We incorporated both, more numerical results, and their meaning in the context of understanding the interaction of geomorphic features and the effect of the sediment concentration.

*Pg3 line 24 to pg4 line 4: shorten this; it feels repetitive. Its basically two outlines/roadmaps of the paper back to back.*
**Response:**

We simplified this paragraph by briefly describing the content of each section of the manuscript.

*Pg 5 line 23: suggest changing to ". . .we do not consider erosion or deposition of the bed."*
**Response:**

We have incorporated the sediment deposition on this statement.

*Pg 5 line 6: Unclear to me if it is the model validation what you show in Appendix A, or is it previous work that needs to be cited, like Guerra et al. 2014? I think the answer is*

*Appendix A; I suggest rewording a bit to make it clear that you present the validation in Appendix A.*
**Response:**

Appendix A details the derivation of the equations for the numerical scheme used in our model. We have modified the text to explain the content of the validation in Appendix A.

*Pg 5 line 11: here and elsewhere, suggest cutting "for details" from citations. Its not needed.*
**Response:**

We have deleted them.

*Pg 7 line 26: this is my ignorance, but I don't know the difference between turbulent and dispersive stresses. It would be helpful to readers like me to work in a 1 or 2 sentence explanation. I see that you address this a bit on the next page (and cite Julien and Paris 2010), but I still suggest a little more.*
**Response:**

We would like to sincerely thank the Reviewer for this comment, since we agree this will make the manuscript more understandable to a wider range of readers.
Turbulent stresses, as its name indicates, are the momentum losses produced by the irregular velocity fluctuations and vorticity that mix the flow and mobilize sediment particles. This is the component mainly associated with the characteristics of the flow that are often described by the Reynolds number.
On the other hand, dispersive stresses are the momentum losses due to collisions of sediment particles. While they both are related, at highly turbulent flows and low sediment concentrations, many particles would remain in suspension, but the dispersive effects would not be significant, since they require large concentrations or large particles that would collide continuously with each other due to the turbulent fluctuations.
Since the references we cite in the manuscript have a clear explanation of each case, we did not include additional ones. However, we have added an explanation in the text to describe the differences.

*Pg 8 line 20: suggest changing "sediment concentrations" to "volume sediment concentrations"*
**Response:**

We have made this change and now we specify that we mean by volume sediment concentration.

*Figure 1: I suggest somehow indicating which channel here is the mainsteam and which is the Quillayes tributary. I can guess or probably figure it out from figure 3, but would be helpful to have more obvious on this figure. Also, in the caption it's unclear to me which*

*black line you're talking about. It looks to me like there's a thin black line around the gridded part, and a thicker black line around the entire watershed.*
**Response:**

We changed the color of the line showing Los Quillayes as turquoise, to distinguish the blue line representing the Quebrada de Ramon. We have added a comment in the caption of the Figure. We have also modified the figure by deleting the black line around the entire watershed and thickening the line surrounding the Lidar covered area.

*Figure 2: while I believe the figure shows a confluence, I can't actually figure out where another stream comes in. Maybe annotate on the photo where the other channel is? Or use a different picture showing it more clearly? I do like the action shows with at least one of the authors for scale.*
**Response:**
We have modified the picture by including two arrows representing the directions of the tributaries.

*Figure 4: suggest changing y axis to Discharge rather than Flow*
**Response:**
We have modified the figure according to this suggestion.

*Pg 11 line 8-9: If CFL was defined earlier I missed it; make sure to define it.*
**Response:**

We have incorporated the definition of the CFL and the value employed in the simulations

*Table 4.1: Some percentages are given in the text, but I suggest just adding columns of "% change from clear water" or similar to the table. I'm surprised at how much difference sed conc makes and think that showing percentages (and editing the abstract) would emphasize this more.*
**Response:**
We have incorporated this values for both streams in Table 1.

*Pg 12 line 10: I'm hesitant to say its an exponential decrease unless there's a plot or other curve fitting showing that an exponential really does work well. Not everything that changes magnitude with time follows exponential decay.*
**Response:**

We sincerely thank the Reviewer for this comment, since our original sentence was not scientifically rigorous. After plotting the variation of the mean velocity of the wave speed against the sediment concentration, we observe that a quadratic function fits better for both rivers. We have corrected the sentence in the manuscript including the $R^2$ of the fitting curve.

[Figure]

*Pg 13 line 3 (at least as the line #s show up on my pdf; its actually farther down the page):*
*I disagree that figure 6 shows "similar dynamics" to figure 5. I'm confused by this. Figure*
*6 is practically inverted from Figure 5. In figure 6, the clear water case propagates slowest,*
*60% propagates fastest (if I'm not confused), which is opposite.*
*There's a bit of possible explanation at the bottom of the page (it has something to do with*
*different arrival times of flood waves on tributaries?), but I don't really understand. I think*
*change "similar dynamics" and explain more what causes the differences between these*
*figures.*
**Response:**

We have modified the two paragraphs that describe Figure 6. We have explained that we observe
the effects of sediment concentrations and topography in both, upstream and downstream the
confluence. We also describe that there is a third factor affecting the dynamics of the flood
downstream of the confluence, which is the difference on the arrival times of the flows coming
from the two streams.

Since the differences of sediment concentration change the flow velocity upstream from the
confluence in different proportions for both streams, we observe that flows with 60% of sediment
concentration arrive at the confluence with a difference of only one hour. However, this difference
increases up to three hours for clear water conditions. As a consequence, the main channel is
flooded due to the volume coming from both, QR and Qui, when the sediment concentration is
60%. For clear water, however, there is a long section of the river that is initially flooded with water
coming only from QR. After the flow advances 4 km downstream from the confluence, it is
reached by the flow coming from Qui, increasing the total flow velocity as shown in Figure 6.

*Pg 15 line 16 (I think line #s are messed up on the version I'm reading; it's a couple lines above figure 8): change "around of 7 h" to "around 7 h"*
**Response:**

Corrected.

*Pg 15 line 25: change "maximum increments of flow depth" to "maximum flow depth"? I may be misinterpreting, but I don't think you're talking about 3 m depth increments, I think you're talking about total flood depths at a given location of ~3m.*
**Response:**

Your assumption is correct, and we apologize for this mistake. We have corrected this error on the new version of the manuscript.

*Figure 10 and 11: Change Q_U to Qui_U, as is used elsewhere in manuscript.*
**Response:**

We updated Figure 7, 8, 10, and 11 to keep a consistent notation.

*Pg 20 line 26: change sensibility to sensitivity?*
**Response:**

Corrected.

*Pg 21 line 20: change extension to extent.*
**Response:**

Corrected.

*Figure B1 caption: change "used to the quiescent. . ." to "used for the quiesce nt. . ."*
**Response:**

Corrected.

*Figure B3: Unclear to me where the dam is located. I suggest adding more explanation to the caption, to explain that the "dams" are between rho1 and rho2? Also suggest in the caption saying that h is the flume width; I was confused a bit about h vs w.*
**Response:**

We have incorporated an explanation of the numerical experiment, and the meaning of h and W as well.

---

## Author Comment (AC2) · 15 Nov 2019

*Response to Referee #2*

We wish to thank the Reviewer for his/her thorough review of our manuscript and for the useful comments that helped us improve the quality of the paper. The specific issues raised by this Referee are addressed in detail below:

*The objectives are not quite clear. We all understand the emphasis on the model and the effects of sediment concentration, but what is it that you are particularly wanting the reader to learn from your research? The model and color figures are nice, but there must be some scientific objective that you tried to accomplish and want to share with the readers.*
**Response:**
We thank the Reviewer for this comment. As we explained in the introduction of the original version of the manuscript:
"*The main objective of this investigation is to gain fundamental insights on the effects of high sediment concentrations on the propagation of floods in an Andean watershed.*"

In the new version of the manuscript we have added additional comments, regarding the competing mechanisms that control the flood dynamics in mountain regions, namely the geomorphic characteristics of the channel, and the rheological effects of sediment concentration.

*The conclusions are long and a bit vague. There should be a clear delineation of what can be concluded from this analysis. Also, the wishful thinking at the end of what you want to do in the future should be left out. There should be a greater emphasis on what has been done and what can clearly be demonstrated from your analysis. What can be achieved in the future should be left out for your next paper…*
**Response:**

We have modified significantly the conclusions to consider this comment. In the new version of the manuscript the Conclusions are shorter and simplified. The main modifications include deleting all the comments referred to future work. We have summarized the paragraphs, simplifying the explanation of how we carried out this research, and most importantly, we have now highlighted the main findings of our research, justifying them with observations obtained from the analysis of our simulations.

*The article is a bit long and there is quite a bit of excess verbiage (a good 10-15% can be trimmed out) that could be deleted without changing the technical content of your discussion. Also, once the paper is approved for publication, it seems better not to include the Appendix in this paper. This material can be useful to the reviewers at this stage of the review process, but will not be necessary in the final paper.*

**Response:**

We have worked on reducing the size of the manuscript, eliminating repetitive content and reducing some paragraphs. We are open to eliminate the Appendix if the Reviewer and the Editor believe that it could make the paper more accessible to the readers.

*The analysis of the effects of sediment concentration is interesting, but the results at a 60% concentration seem too fluid and flowing quite fast. Depending on the amount of clay and the type of clays, the flows at such a concentration can be very different than modeled. These hyper-concentrated flows may also resembling very slow moving mud flows. It may also be useful to indicate whether this is a concentration by weight or by volume. It does make a large difference at high concentrations.*
**Response:**

*We thank the Reviewer for this comment. We are aware that many factors control the complex dynamics of hyperconcentrated flows, which determine their rheological behavior and the velocity of the floods. (Julien and Leon, 2000).*

*The nature of the flow highly depends on the characteristics of the sediment particles, and specifically their size and composition. Slow moving flows typically occur in flows with high concentrations of silts and clays (fine sediment sizes). In these cases, the turbulent and dispersive stresses lose importance, and the yield and viscous stresses control the dynamics of the flow (Widjaja and Hsien-Heng Lee, 2013). Additionally, the cohesion between particles plays an important role on the flow resistance.*

*In the cases we are analyzing, however, the sediment is generated in a section of the Andes where the smallest fractions of sediment correspond to fine sand, and almost no clay is present. **We are therefore considering particles with diameters that vary from 2 to 10 mm, which is considered very fine sand to medium size gravel** (Julien, 2010).*

*In our simulations, the flow is a mixture of clastic material and a lubricating fluid and the main mechanisms for energy dissipation are collisions among particles (dispersive stresses) and turbulence at high flow velocities.*

*Finally, we would like to underscore that the quadratic rheological model used in this work considers all the stresses previously mentioned. Depending on the characteristics of the flow and sediment, and the terms representing each effect can take high or low values of the stresses.*

*As we explain in the new version of the manuscript, all the concentration values are expressed in terms of volumetric sediment concentration.*